# Combined impacts of deforestation and wildlife trade on tropical biodiversity are severely underestimated

William S. Symes [1], David P. Edwards[2], Jukka Miettinen[3], Frank E. Rheindt [1] & L. Roman Carrasco [1]

Tropical forest diversity is simultaneously threatened by habitat loss and exploitation for wildlife trade. Quantitative conservation assessments have previously considered these threats separately, yet their impacts frequently act together. We integrate forest extent maps in 2000 and 2015 with a method of quantifying exploitation pressure based upon a species' commercial value and forest accessibility. We do so for 308 forest-dependent bird species, of which 77 are commercially traded, in the Southeast Asian biodiversity hotspot of Sundaland. We find 89% (274) of species experienced average habitat losses of 16% and estimate exploitation led to mean population declines of 37%. Assessing the combined impacts of deforestation and exploitation indicates the average losses of exploited species are much higher (54%), nearly doubling the regionally endemic species (from 27 to 51) threatened with extinction that should be IUCN Red Listed. Combined assessment of major threats is vital to accurately quantify biodiversity loss.

[1] Department of Biological Sciences, National University of Singapore, 14 Science Drive 4, Singapore 117543, Singapore. [2] Department of Animal and Plant Sciences, University of Sheffield, Sheffield S10 2TN, UK. [3] Centre for Remote Imaging, Sensing and Processing (CRISP), National University of Singapore (NUS), 10 Lower Kent Ridge Road, Singapore 119076, Singapore. Correspondence and requests for materials should be addressed to W.S.S. (email: wsymes@u.nus.edu)

Tropical forests are the most biodiverse ecosystem globally[1]. In recent decades, there has been an extensive loss-of-tropical forests, driven primarily by the expansion of agricultural land[2–4]. This loss has serious consequences for tropical biodiversity, as the destruction of suitable habitat threatens the survival of forest specialist species.

Anthropogenic disturbances within remaining forest, including logging, fires, hunting, trapping, and edge effects, are also serious drivers of biodiversity declines[5]. Hunting is now a major cause of biotic population declines across the tropics[6], and, in hunted forests, has caused declines of 58 and 83% in bird and mammal populations, respectively[7]. Illegal hunting of wildlife for internationally traded products, pets and as a food resource are directly responsible for the declines of emblematic species, such as elephant[8], rhinoceros[9], tiger[10], and Bali starling[11]. At its most extreme, overhunting can result in the extinction of large-bodied animals in otherwise healthy intact habitat[12], driving changes in forest composition[13–15].

The negative consequences of habitat change and wildlife exploitation are thus often cumulative. Increasing deforestation and forest fragmentation and its associated infrastructure development makes remote areas of forest increasingly accessible[7,16], exacerbating hunting and trapping pressure and other forms of anthropogenic disturbance[5]. Despite the well-established links between hunting/trapping, accessibility, and forest fragmentation, to date, most quantitative conservation assessments of extinction threat fail to account for their compounded impact, focusing on the impacts of deforestation[17,18] or hunting[7,19,20] in isolation. While recent attempts to incorporate up-to-date spatial data globally[18] represent an advance in our understanding of species threat assessments, by looking at one type of threat in isolation, they likely underestimate the extinction threat posed by the combined impacts of wildlife trade and deforestation in commercially valuable species.

We focus on Sundaland, a major hotspot of biodiversity in Southeast Asia, where habitat loss, hunting and wildlife trade are particularly acute[9]. Seventy percent of original forest cover was lost by 2010 with projected devastating consequences for the region's biodiversity[12,21], and the expansion of industrial oil palm and paper-pulp plantations is on-going[22]. These developments are compounded by an intensive wildlife trade for birds and other species that feeds rampant domestic and international markets, driving precipitous population declines and local extinctions in many species[11,23–25]. Focusing on 308 forest-dependent bird species, 77 of which are heavily trapped for pets, products or as a local food resource, we show that the combined impacts of deforestation and exploitation makes the declines in exploited species much higher than previously thought.

## Results

**Loss-of-forest habitat**. We found virtually all lowland forest specialist birds (274; 89%) experienced loss of suitable habitat (ESH) in Sundaland (which contains on average 75% of the global range of our study species) between 2000 and 2015 (mean loss = 14.7% ± 9.7 standard deviations (SD); Fig. 1). The maximum loss was 39.1% [Sumatran babbler *Trichastoma buettikoferi*; see Supplementary Data 1 and Supplementary Data 2 for full details]. Deforestation thus remains a severe conservation threat (Supplementary Figure 1). Using a simple reverse species-area relationship our analysis suggests between 16.9% (52 species) and 30.1% (92) of all forest-dependent species will go extinct in the region by 2100 (Supplementary Figure 2).

**Impacts of species exploitation**. We identified 77 species as being commercially valuable. We first determined how much of each of these species' range occurred within 5 km of a forest edge, given that hunters and bird trappers are known to travel at least this distance on foot from roads, but usually further[7,26]. We then used a combination of published information and expert opinion to classify species into persecution and thus exploitation pressure categories, identifying 24, 24, and 29 species suffering high, medium, and low persecution, respectively. The majority of traded species' ranges were within 5 km of a forest edge (mean = 82% ± 9.3 SD) (Fig. 2), and we found 10 species that have over 99% of their range within 5 km of a forest edge, including Silvery Pigeon (*Columba argentina*) and Nias Hill Myna (*Gracula robusta*). For species not classified as exploited (230 species) we conservatively assumed the impact of exploitation to be 0.

Integrating persecution with proximity to edge (adjusted for the proportion of the range in Indonesia), we estimated exploitation would be responsible for a mean population decline of 36.6% (±24.6 SD) across all commercially valuable species, equating to 17.1% in low, 30.7% in medium, and 66.1% in high persecution categories (Fig. 3). We estimated the impact of exploitation to be particularly high in species endemic to Java, with Melodious Bulbul (*Alophioxus bres*), Javan Leafbird (*Chloropsis cochinchinensis*), and Javan Hawk-eagle (*Nisaetus bartelsi*) all experiencing population declines above 90% (for a full breakdown see SI).

**Combined declines from deforestation and exploitation**. We estimate the combined declines from deforestation and exploitation to be 23.9% (±21.6 SD) across all forest-dependent (exploited and non-exploited) species in the region. The average decline for the 77 exploited species was 15.3% (±10.1 SD) from deforestation alone, rising to 51.9% (±23.2 SD) decline when deforestation and exploitation are combined. When comparing losses from deforestation to those from exploitation, the latter was a more pressing cause of decline than habitat loss for 58 (75%) commercially valuable species, including 12 (41%) species that are only persecuted at low levels, 22 (91%) medium-persecuted, and 24 (100%) high-persecuted species. This underscores the critical importance of quantitative assessments that combine deforestation and estimates of exploitation pressure, and illustrates that we are likely to be seriously under-estimating the extinction risk of traded species.

**IUCN Red List assessment**. Informed by our assessment of deforestation and exploitation impacts, we conducted an IUCN Red List assessment for the 202 regionally endemic species (defined as having ≥80% of their range in Sundaland). We estimated declines in habitat over 3 generations or 10 years since 2000[27]. Based only on deforestation, we found 11 species should be listed as Endangered (EN; 4 species) or Vulnerable (VU; 7), whereas based only on exploitation, we estimate 28 species should be listed as Critically Endangered (CR; 5), EN (10), or VU (13).

When we combined the impacts of deforestation and exploitation, our results suggest that a total of 51 species should be listed as CR (9), EN (20), or VU (22) (see Supplementary Data 1 and Supplementary Figure 3), which represents an 89% increase from the 27 species currently listed by the IUCN, and with most of this increase in the CR (+5, from 4) and EN (+15, from 5) categories. Notably, new CR species included the currently least concern (LC) Melodious Bulbul (*Alophioxus bres*), while only seven CR and EN species—Black Hornbill (*Anthracoceros malayanus*), Barred Eagle-owl (*Bubo sumatranus*), Long-tailed Parakeet (*Psittacula longicauda*), Rhinoceros Hornbill (*Buceros rhinoceros*), Storm's Stork (*Ciconia stormi*), Wallaces's Hawk-eagle (*Nisaetus nanus*) and Wrinkled Hornbill (*Rhabdotorrhinus corrugatus*)—had habitat loss as the primary cause of

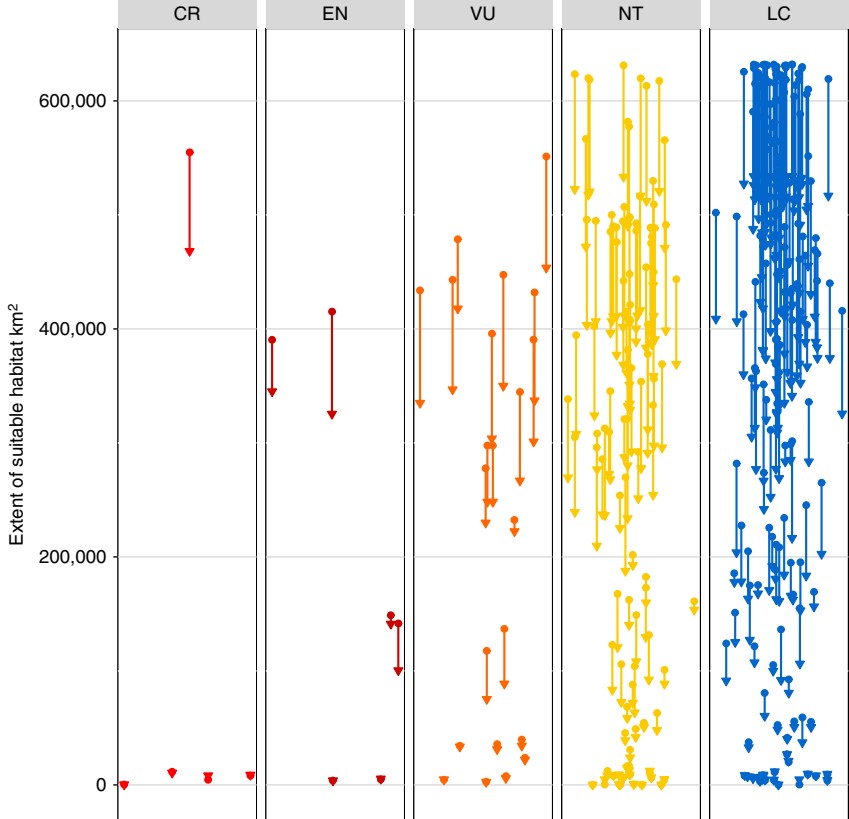

**Fig. 1** Loss-of-habitat in forest-dependent Sundaland birds. Change in suitable habitat, not including the impact of exploitation, for each of the 308 studied species between 2000 and 2015, split by their current IUCN Red-list status: critically endangered (CR), endangered (EN), vulnerable (VU), near threatened (NT), and least concern (LC). The circles represent the extent of suitable habitat in 2000 and the triangles in 2015; the lines are drawn between the circle and triangle for the same species to highlight the species-specific change

population decline, with exploitation the leading driver in the remaining 22 species (Fig. 4). We also found that four species, all Enggano endemics, meet the criteria for Endangered based on having an extent of occurrence less than 500 km².

We calculated declines for several species that would not meet the thresholds for their current IUCN status based on decline-related criteria, indicating that they could be downlisted. Most notably, these include Javan Blue-banded Kingfisher (*Alcedo euryzona*) from CR to LC, Silvery Pigeon (*Columba argentina*) CR to EN, White-rumped Woodpecker (*Meiglyptes tristis*) EN to LC, and Straw-headed Bulbul (*Pycnonotus zeylanicus*) EN to VU. However, for some of these species, causes of endangerment are not covered appropriately by our methodology so that we do not recommend their downlisting based on our results (Discussion). We also estimated declines of below 30% for seven species currently listed as VU (see Supplementary Data 1 for details). For some of these species, our method may have underestimated threats, especially when exploitation occurs outside our geographical extent of analysis (i.e., in Malaysia).

**Coverage by protected areas**. We assessed the amount of coverage that species have within protected areas (PAs) and found that the majority of regionally endemic species (175) had more than 2000 km² (the IUCN area of occupancy threshold for Vulnerable status) of suitable habitat within a PA. A mean of 16.4% (±6.7 SD) of species' ranges were within PAs (Fig. 5), but there was substantial variation with up to 38.2% of a species range protected [Cream-striped Bulbul (*Pycnonotus leucogrammicus*)) and with four species having none of their range protected (Simeulue Scop-owl, (*Otus umbra*); Simeulue Parrot (*Psittinus*

*abbotti*); Silvery Pigeon (*Columba argentina*); Nias Hill Myna (*Gracula robusta*)]. There were no large differences between the percentage of species' ranges within a PA for each IUCN Red-list status: 21.5% for CR, 15.4% EN, 15.7% VU, 16.1% NT, and 17.9% LC.

Although protected areas can prevent deforestation (but see Brun et al.[28]), there is a substantial risk that they do not prevent exploitation[26]. We found on average only 5.6% (±3.4 SD) of the species' extent of suitable habitat was both protected and beyond 5 km from a forest edge. Twenty-three persecuted species then fell below the 2000 km² threshold for VU, eight of which have less than 10 km² of their range within PAs that is not susceptible to hunting, the threshold for CR.

## Discussion

Our study highlights the importance of considering the impacts of major conservation threats in combination: recent habitat loss and exploitation combine to drive dramatic extinction risks to the forest specialist species of Sundaland. Without urgent policy intervention to curb deforestation and slow the quantities of birds entering the cagebird trade, many species are likely to be lost. Failing to account for these combined threats can lead to a major underestimation of threats in Red List assessments.

Our analysis suggests that exploitation for wildlife trade has caused dramatic declines in many species within the region and underscores the critical role that effectively guarded PAs could play as reservoirs of these species. It remains poorly unknown, however, whether PAs are effective at reducing bird exploitation on the ground and future research should point in this direction. Several factors suggest the impacts of exploitation will continue,

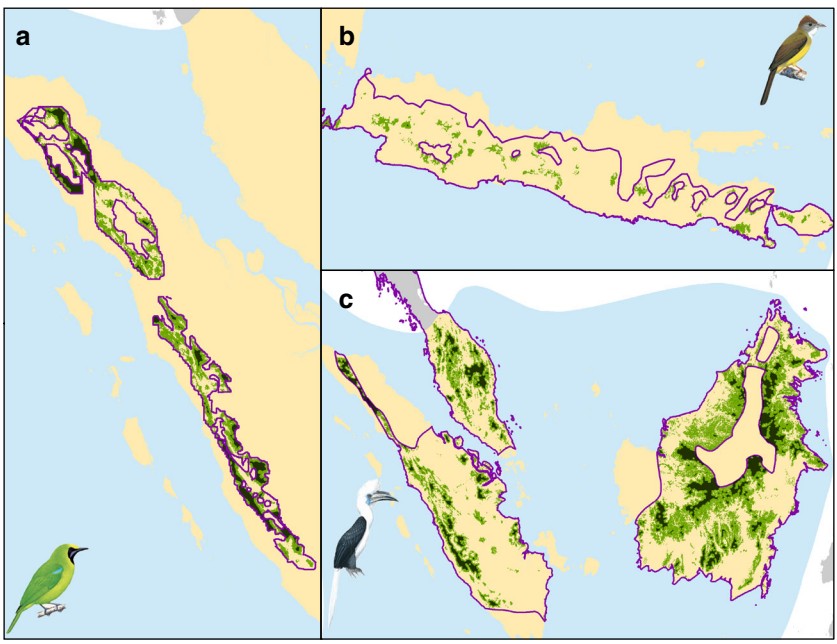

**Fig. 2** Illustration of the mapping methodologies showing the impacts of habitat loss and hunting. The three panels illustrate the ranges accessible to trappers for three species in our analysis: **a** Sumatran Leafbird (*Chloropsis media*), **b** Melodious Bulbul (*Alophoixus bres*) and **c** White-crowned Hornbill (*Berenicornis comatus*). The purple line is the outline of the species' historic range (as provided by BirdLife International). The green area, which is divided into two shades, indicates the total extent of suitable habitat for the species in 2015, once it has been refined for current forest extent and elevation. The dark green regions are areas that are further than 5 km from the forest edge and considered inaccessible to trappers; the light green areas are regions that are within 5 km of a forest edge where exploitation is likely taking place. Species illustrations are not within the CC-BY license of this publication, and instead are reproduced from del Hoyo, J., Elliott, A., Sargatal, J., Christie, D.A. & de Juana, E. (eds.) (2018). Handbook of the Birds of the World Alive. Lynx Edicions, Barcelona. (retrieved from http://www.hbw.com/ on [23/08/2018]). All rights reserved. Basemap: © EuroGeographics for the administrative boundaries

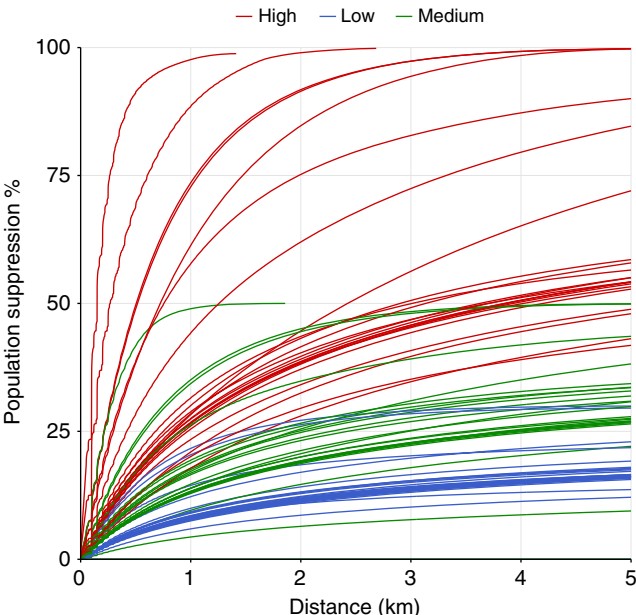

**Fig. 3** Hunting impacts for each of the 77 commercially valuable forest bird species in Sundaland. Each line represents a species and shows the cumulative expected population suppression by exploitation between 0 km and 5 km from forest edge. Lines are coloured by persecution category

including within PAs. First, as popular species become rarer their commercial value will likely increase, accelerating exploitation efforts and declines, potentially resulting in an anthropogenic Allee effect[29]. Further, as rare species drop out of the market,

replacements are often substituted into the market [e.g., the recent emergence of Greater Green Leafbird (*Chloropsis sonnerati*)[25]]. Second, unpredictable responses to cultural phenomena can result in previously unexploited species becoming the target of trappers: in Indonesia, the recent popularity of owls is dubbed the 'Harry Potter effect'[30]. Third, on-going fragmentation via deforestation and road development into contiguous forest[16] (especially in Borneo) will make forest interiors increasingly accessible to trappers, further reducing the number of isolated refugia for commercially valuable species. Recent research suggests that the level of unmapped roads is very high, pointing to increasingly high accessibility of forests for exploitation[31]. Finally, a lack of funding (annual shortfall of US $521 million per year in Indonesia) for patrols and insufficient law enforcement and punishment of exploitation means that many PAs do not effectively prevent trapping and hunting[7,19,26,32]. This is particularly concerning given that all of the regionally endemic, persecuted species have below 10% of their range within the core of a PA (i.e., >5 km from an edge). Thus, PAs will likely only protect the subset of non-persecuted species by potentially reducing habitat loss and they may fail to prevent extinctions of many commercially valuable species.

Our study also underscores the importance of deforestation as an extinction driver. Forest loss, largely due to the expansion of agriculture, has a direct negative impact on the majority of forest species within the region, and this reduction is on-going in Sumatra, Borneo, and Peninsular Malaysia[33,34].

Previous analyses have used a reverse species-area relationship to predict the direct impact of deforestation on extinction risk in the region, estimating that between 24 and 42% of all biodiversity faces extinction[12,35] (but see He & Hubbell[36]). Our results are somewhat lower [16.9% (52 species)—30.1% (92)] likely reflecting

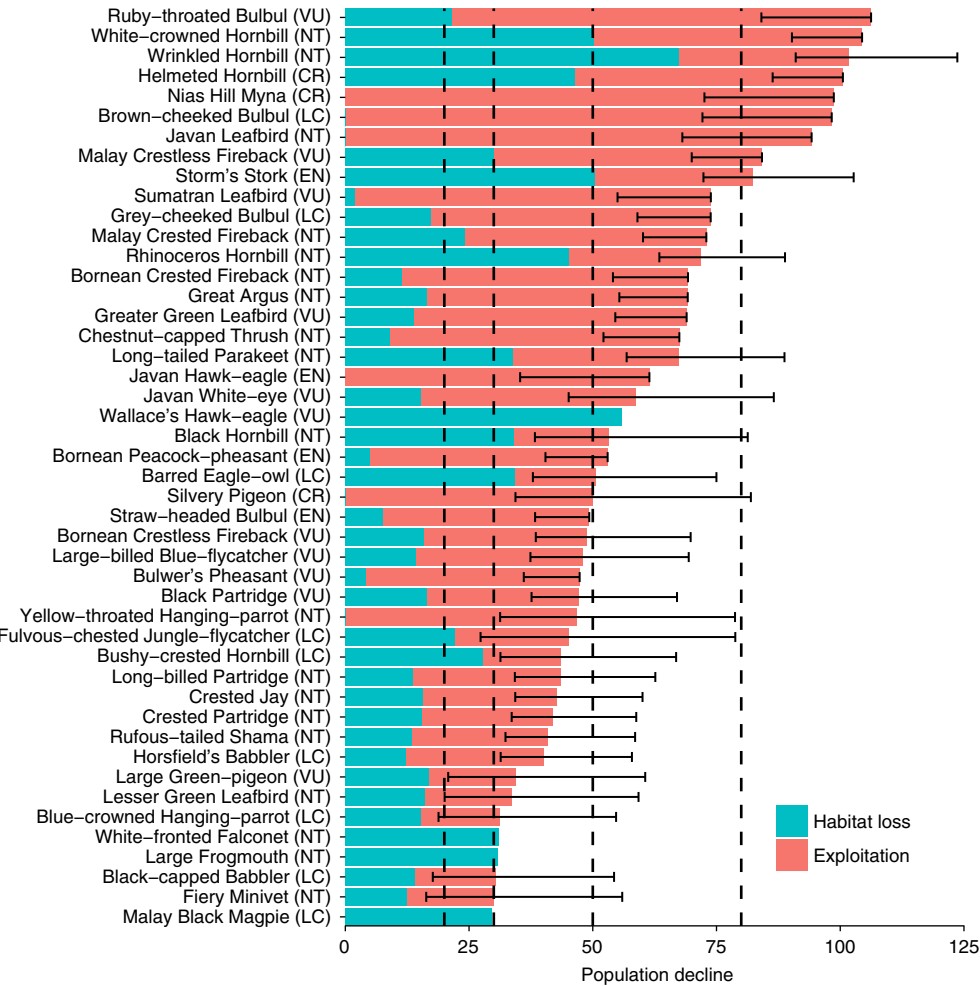

**Fig. 4** Combined population declines from habitat loss and exploitation. The blue bar is the contribution of habitat loss and the red bar the contribution of exploitation. This graph only shows the 45 regionally endemic species with a total predicted population decline of over 30% (for all species see supplementary online material). Error bars represent the estimation uncertainty of population decline due to exploitation, they are calculated using the 2.5% and 97.5% intervals of a PERT distribution. Vertical lines represent the thresholds for classification as near threatened (NT) (20%), vulnerable (VU) (30%), endangered (EN) (50%), and critically endangered (CR) (80%). Values above 100% result from adding the effects of habitat loss and exploitation and are interpreted as population declines of 100%

an increase in the accuracy of remote sensing analyses from previous studies. However, our figures do not include the additional extinction risks posed by hunting and trading of commercially valuable species (Supplementary Figure 3).

While considerable debate surrounds the actual rate of extinctions solely from habitat loss[36,37], in combination, habitat loss, and hunting have resulted in numerous extinctions globally at both the island (e.g., Mascerenes[38]) and continental (e.g., megafaunal extinctions in the Late Quaternary[39]) scales. Given the particularly acute nature of these threats in Sundaland, without concerted conservation efforts to greatly reduce deforestation and exploitation, the region is at significant risk of being an extinction hotspot in the future.

While our method is a rapid and straightforward way for assessing population declines, there are several limitations that must be noted. In assessing the impact of exploitation, detailed information on the behaviour of wild bird trappers and species responses to exploitation is not available. Hence, we made simplified assumptions regarding hunting impact and accessibility, but in reality, species responses to hunting are more nuanced than the three categories we used (low, medium, and high)[40] and accessibility is a complex interaction of population, roads, topography, and markets. We attempted to account for this

uncertainty by combining maps of major roads and all available roads to calculate additional metrics of accessibility. Using road maps instead of forest edges made considerable difference to our results, with both road maps leading to considerably lower threat estimates.

Using only major roads only 16 species were above Red List thresholds (6 EN and 10VU) and with all roads 38 (3 CR, 13 EN, 22 VU) (Supplementary Figure 4 and Supplementary Figure 5). However, the maps we used, while the best available (Open Street Map and WRI produced Indonesia map), are fundamentally inaccurate with many roads missing[31,41]. Crucially, this inaccuracy is not uniform and changes the analysis in biased and unpredictable ways. For example, the maps contain no roads inside protected areas in Java, suggesting Javanese forests are much more isolated than they actually are, leading to large underestimations of threat. Second, deciding to what extent individual roads are accessible to hunters requires further assumptions. By using distance from forest edge we avoid using maps that are known to be inaccurate[41] and we can account for access from other means (such as rivers) by assuming deforestation follows these points of access.

Because our knowledge of trade dynamics and trapper behaviour is best in Indonesia, we made the conservative assumption

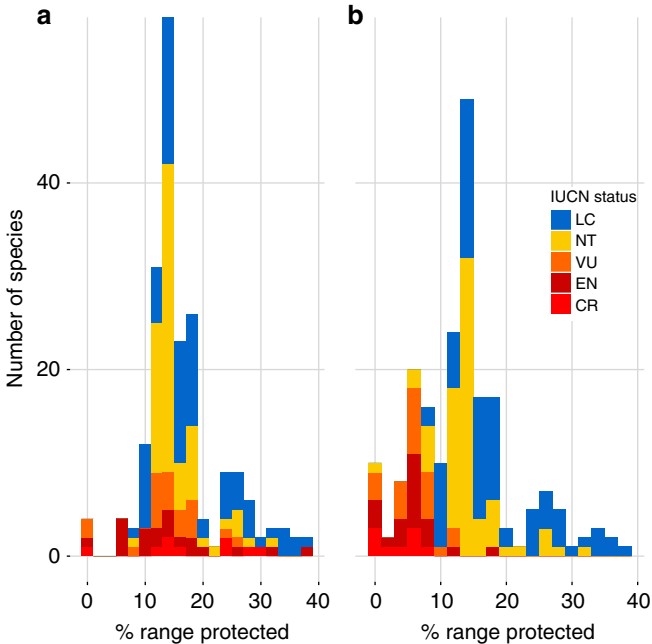

**Fig. 5** Area of suitable habitat inside protected areas. Histogram of the number of regionally endemic forest-dependent bird species (77 in total) against percentage of a species' range that falls within a protected area (PA; IUCN category I–V); bar colour represents IUCN status based on our analysis. **a** The percentage of the range within PAs, and **b** the percentage of the range protected within PAs once exploitation-susceptible areas (i.e., within 5 km of a forest edge) have been removed

that there was *no* exploitation outside Indonesia, but this is certainly not the case for many species[24,25] and—at worst—led to unrealistic estimates in globally endangered species that are either highly persecuted outside of Indonesia (e.g., Straw-headed Bulbul, *Pycnonotus zeylanicus*[25]) and/or have the majority of their range in Malaysia (e.g. Malay Peacock-pheasant, *Polyplectron malacense*). Despite these limitations, our results agree with a recent meta-analysis of hunting, which showed global average population declines of 58% (versus 36.6% in this study), and depletion within 7 km of access points, further highlighting the role of the pet trade in driving defaunation[7,42].

In predicting losses to deforestation, the underlying maps, while the best available, likely contain commission and omission errors (e.g., areas that are either included or excluded erroneously), which could lead to inaccurate decline or exploitation estimates. The changes in mapping methodology outlined in the methods also led to conservative/underestimates of ESH decline, at its most extreme 23 species (Javanese endemics and range restricted Island species) experienced an increase in ESH between 2000 and 2015. However, all the species affected are restricted to extremely limited extents and hence, have likely remained largely unchanged in the analysis period. We therefore do not expect any substantial population increases in these species, instead the increases highlight how our estimated decreases in other species are likely conservative.

Also, inaccuracies may occur where species' habitat requirements are not fully understood [e.g., Bonaparte's Nightjar (*Caprimulgus concretus*)]. Our method is not sensitive to species that have very specialised habitat requirements, or exist at very low densities within suitable habitat. For example, the Javan Blue-banded Kingfisher (*Alcedo euryzona*), which is limited to lowland and hill riverine forest that is not specifically defined on our maps, is suggested to be downgraded from CR to LC in our analysis, but riverine forest has been particularly affected by

deforestation in the region[43]. Consequently, while our results represent an improvement on existing knowledge in the majority of cases, each assessment must be judged in context, as would occur in any normal IUCN assessment process.

We also assume a linear relationship between deforestation and population decline, which we believe is a conservative assumption given the negative impacts that edge, area and isolation effects have on species[44]. There is considerable debate as to whether the relationship is linear, with many theoretical (and some field) studies suggesting that populations can remain reasonably stable until a certain threshold of habitat loss (for a comprehensive review see Swift and Hannon[45]), leading to overestimates of population decline under linear assumptions. However, while habitat thresholds may exist for some species in our study, given our current knowledge, calculating these thresholds accurately is impossible and as such they cannot be incorporated into our analysis and would be of questionable utility to conservation decisions[45]. Instead, our assumption of linearity is in line with the precautionary principle (which is acknowledged in the Convention of Biological Diversity), since it ensures we are not underestimating declines by assuming a threshold that does not exist, and thus is more useful to conservation decisions. For simplicity, we also assumed that the effects of the loss of habitat from deforestation and exploitation were additive, which results in three species having maximum estimated declines above 100%. In reality, the proportion of the remaining area of a species subjected to exploitation will increase as the habitat is reduced and fragmented, meaning the impacts are likely synergistic[46], making our estimates for most species conservative.

Our results suggest that by failing to account for the combined impacts of habitat loss and exploitation, the Red List currently underestimates the threats facing many species. By incorporating quantitative measurements of habitat loss and exploitation, our Red List assessments differed substantially from the current IUCN status. Currently, only 27 species in the region are Red listed (VU, EN, or CR), whereas our results indicate that this should increase by more than 80% to 51 species, but only if deforestation and exploitation threats are considered together. By incorporating exploitation impacts within spatial assessments of edge effects, we have identified species that are likely suffering precipitous, undocumented population declines.

In this study, we estimated the rates of population decline over 3 generations or 10 years, for assessment under the IUCN criteria A3 using an index of abundance and actual levels of persecution. The mode of assessment applied can make a significant difference to the end result. In our case, criteria E (a quantitative assessment of extinction risk) could be used via a species-area relationship calculation, but we chose not to use this method because the SAR has been previously criticised for over-estimating extinction risk from habitat loss[36]. However, future reassessments using this criterion and more complicated extinction-risk models are a valuable area for future research.

While we assessed two major threats in combination, we did not consider other threats, such as the impacts of logging, which will likely cause substantial reductions in some species. In Malaysian Borneo, for instance, 92 of our study species suffered from reduced abundance following intensive selective logging[47]. Given the vast majority of lowland forest remaining in the region has been selectively logged[48,], our Red List assessments are probably conservative for many species. We did not incorporate logging effects into our assessment because the impacts on many species remain unknown (especially those restricted to Java and Sumatra). The same is also true for the impacts of increasing fragmentation, which we do not directly consider, but are likely to have profound implications for many species[44]. Incorporating logging impacts and fragmentation effects represents another

important frontier in combined species assessment, especially in logging ravaged regions in the tropics (i.e., Southeast Asia, Congo, Southern, and Eastern Amazon).

Another key issue, with profound implications for conservation designation, is where the limits are drawn for full species status. For example, there is controversy surrounding the taxonomic treatment used by the IUCN Red List for birds[49]. Five recently suggested elevations of sub-species to full species status[50] in the region are not recognised by the IUCN, with at least one species qualifying as CR [Barusan Shama (*Copsychus melanurus*)), one as EN (Enggano Parakeet (*Psittacula modesta*)) and another as VU (Brown Wood Owl (*Strix indranee*)] by our estimation. This discrepancy emphasizes how the blanket application of a single taxonomic treatment can exacerbate the underestimation of threats facing biodiversity in Red List assessments[50,51], suggesting that conservation would benefit from mechanisms to assess extinction risk under alternative taxonomic treatments on the IUCN Red List.

In conclusion, our results uniquely highlight the precipitous declines of many Sundaic forest-dependent birds over the last 15 years from the combined impacts of rapid deforestation and exploitation and, as a result, that current IUCN Red List assessments underestimate threats in the region. For commercially valuable species, wildlife trade is the leading cause of decline in the majority of cases, yet very little information is available on the dynamics of trade, the behaviour of trappers, and in turn, population responses of traded species, indicating an urgent research need[26,52]. While a slowing of deforestation is essential to limit extinctions of forest-dependent birds, without coordinated efforts to curb commercial exploitation, including better protection in PAs and stronger law enforcement, numerous extinctions of commercially valuable species appear inevitable. Finally, the combined impacts of deforestation, forest fragmentation and commercial exploitation are not unique to Southeast Asia; for example, rampant land-use change and wildlife trade drives declines in parrots from Latin America, Africa, and mainland Asia[53,54]. Therefore, the extinction risks from deforestation and exploitation may be severely underestimated globally, making it essential for future quantitative conservation assessments to take into account the combined effects of habitat loss, hunting, and exploitation.

## Methods

**Study region.** Our study region was the biogeographic region of Sundaland that encompasses the western half of the Indo-Malayan archipelago (an area of ~1.6 million km$^2$) and contains around 17,000 equatorial islands, including Borneo, Sumatra, and Java. This global hotspot of biodiversity has high levels of endemism, with 264 endemic (of 796) bird species, 172 (380) mammal species and an estimated 15,000 (25,000) plant species[55,56].

**Habitat loss across bird ranges.** We obtained range maps and ecological data for all species of birds occurring in Sundaland from BirdLife International using the most recent 2017 range maps[43]. We then filtered our list to include only lowland forest specialist species, defined as those species with the majority of their range below 500 m above sea level and those described as being forest-dependent according to BirdLife International[57]. This list was further refined based on expert opinion (F.E.R. and D.P.E.) to remove/add species with inaccurate forest dependency assessments. Our assessment led to the inclusion of four species which had been classified as non-forest dependent by BirdLife: White-crowned Forktail (*Enicurus leschenaultia*), Lesser Fish-eagle (*Icthyophaga humilis*), Grey-headed Fish-eagle (*Icthyphaga icthyaetus*), and Grey-cheeked Green-pigeon (*Treron griseicauda*). We also excluded forest-dependent species that can be found breeding in forest plantations. We adhered to the BirdLife taxonomic treatment to ensure our results and recommendations are policy relevant. After filtering of upland and non-forest species, 308 lowland forest specialist species remained for analysis.

Next, we clipped the species' range maps for forest extent in 2000 and 2015 to estimate the extent of suitable habitat (ESH) within each range. We created our forest extent maps by combining all the primary forest classes (mangrove, peat swamp forest, lowland evergreen forest, lower montane evergreen forest, and upper montane evergreen forest) from land cover maps for 2000 and 2015 taken from

Miettinen et al.[34] and Miettinen et al.[58], respectively. Primary forest includes areas of forest degraded by selective logging, which dominates in the lowlands of Sundaland[59]. The 250 m resolution maps were created with a semi-automated classification approach using Moderate Resolution Imaging Spectroradiometer (MODIS) data[34,58].

There were some methodological differences in the creation of the two maps (outlined in Miettinen et al.[58]), which were taken into account in our analysis. By combining all primary forest classes (which includes logged forests) and limiting our interest only to forest extent, we effectively removed all methodological differences between the two maps, apart from the difference in the level of disturbance allowed in forest areas. In the 2015 map, more disturbance was allowed in the forest classes to minimize the exclusion of selectively logged primary forests. This difference between the mapping approaches could not be removed. This essentially means that the 2015 forest extent includes some disturbed forests which would not have been classified into the primary forest classes at all if the 2000 map methodology had been used. However, this only makes our estimates of the changes in forest extent, and thereby the range losses, more conservative and was therefore not seen as a crucial limitation for the analysis.

We refined our bird range maps by clipping them to reflect the elevational ranges for each bird from BirdLife International and the NASA shuttle radar telemetry digital elevation model with a resolution of 90 m[60]. We did not further refine the range maps of species that had no information regarding minimum and maximum elevation ranges (88 species). We performed all the spatial analysis for this project in the Python 2.7.3 coding environment using tools provided by ArcGIS 10.3.1. Clipped bird range maps were projected into the Behrmann projection (an equal area projection) and their areas calculated. All subsequent analyses were performed in this projection to maintain consistency. We then calculated the range loss for each species between 2000 and 2015 by subtracting the estimated range in 2015 from the estimated range in 2000.

Finally, we were unable to calculate the change in available habitat for 4 species: the Mentawai Scops-owl (*Otus mentawi*), Mentawai Malkoha (*Phaenicophaeus oeneicaudus*), Silvery Pigeon (*Columba argentina*), and Pink-headed Imperial-pigeon (*Ducula rosacea*) owing to a void in our land cover map from 2000 affecting around 70% of the islands in the Mentawai archipelago and other small island groups. For these species and the 4 species of Enggano endemic birds, we calculated the extent of occurrence using a minimum convex hull as suggested by ref. [61]. These analyses were performed using ArcGIS 10.3.

**Exploitation impacts.** To assess the impact of wildlife trade on commercially valuable species, we estimated the accessibility of the bird range to trappers and hunters. Hunting and trapping are fundamentally different processes, with different drivers and often different actors. In Sundaland, we categorised three types of threat: international trade, domestic trade, and local hunting. We classify the 'domestic market' as trade within Indonesia, which is particularly important for song birds and is dominated by cagebird trapping. International trade is important for a small number of commercially valuable species [e.g., Helmeted Hornbill *Rhinoplax vigil*]. Our final category, 'local hunting', is mostly perpetrated by small-scale actors, often opportunistically killing large-bodied species for consumption and/or selling on local markets. For a full breakdown of which species are in which category see Supplementary Data 1. We define exploitation in our study as a combination of all three of these processes since they all result in population declines.

We created a path distance raster (with cell size of 150 m) for each of the range maps. The value of each cell in a path distance raster is the distance from the middle of that cell to the closest edge of the forest taking into account changes in elevation. From this raster, we calculated the percentage of the entire range for the 77 exploited species that was within a given distance from the forest edge, which we used to assess the proportion of the bird species' range in which exploitation could take place. To test the sensitivity of our result to different access points we also created path distance rasters based on the road network in the region obtained from OpenStreetMap (for Malaysia, Singapore, and Brunei)[62] and the Peta Dasar (for Indonesia)[63]. We then calculated the proportion of the species ranges that was both within 5 km of a road and inside forest from the path distance rasters. We repeated this analysis twice, first for only major roads and again including all roads in the maps.

Empirical data on commercial exploitation of birds in South-East Asia are scarce. We therefore followed the results of Harris et al.[26], which found the median maximum distance trappers travel into the forest to be 5 km. Two expert ornithologists from the region (F.E.R. and D.P.E.) separated the commercially valuable species into three categories—high, medium, low—based on their value to trappers, as indicated by published[25,30,64] market surveys or reviews. F.E.R. and D. P.E. assigned an expected, maximum and minimum exploitation pressure (efficacy) at 5 km to these three categories to reflect their uncertainty in the categories given. We used these estimates to parameterize PERT (programme evaluation and review technique) distributions (shape = 4). PERT distributions assign low probabilities to extreme values and thus are useful for assessing the uncertainty in expert estimates. In our case, we assigned a minimum, expected, and maximum exploitation impact estimate and used the 2.5% and 97.5% percentiles as 95% uncertainty ranges. We identified 77 species of forest birds within the region as being targeted by trappers and, of these, 24 were identified as being under high persecution, 24 medium, and 29 low (see Supplementary Data 1 and Supplementary Data 2). We estimated the

exploitation efficacy at 5 km for the three groups as high = 100% (73.7–100, 95% confidence intervals), medium = 50% (34.5–82) and low = 30% (6.9–75.2). Since exploitation effort differs across the countries in the region, with dramatically more trade in Indonesia than Malaysia[64,65], we conservatively assumed exploitation was only an issue in Indonesia and adjusted our estimates of exploitation by the proportion of the range inside Indonesia. We therefore calculated the population reduction due to exploitation using:

$$R_{ih} = P_{i\,indo} \cdot E_{i\,5\,km} \cdot H_{i\,5\,km}, \tag{1}$$

where $R_{ih}$, the proportion of population reduction for species $i$ due to exploitation, is defined as $P_{i\,indo}$, the proportion of the range of bird $i$ in Indonesia, multiplied by $E_{i\,5\,km}$, the proportion of the bird range within 5 km of the edge and $H_{i\,5\,km}$, the exploitation efficacy up to 5 km from the forest edge.

**IUCN Red List assessment**. We performed an IUCN Red List assessment based on the two proxies of population decline we calculated, habitat loss and exploitation. We assumed the rate of habitat loss was directly proportional to the rate of population decline, such that a loss of 1% suitable habitat per year was equivalent to a 1% population decline per year. We used Red List criterion A4 (an observed, estimated, inferred, projected or suspected population reduction where the time period must include both the past and the future (up to a max. of 100 years in future), and where the causes of reduction may not have ceased OR may not be understood OR may not be reversible), using the generation times provided by BirdLife International and the species-specific rate of habitat loss, to calculate the expected population decline over three generations or 10 years (whichever was longer)[27]. For species in which three generations are longer than the 15 year time period observed (60 of 202 endemic species) we assumed forest would continue to be lost at the same observed rate, a plausible and potentially conservative assumption given that the rate of deforestation has been increasing in the region[3]. We then used a proportional decay function to predict the proportion of the species range that would be lost:

$$a_{it} = a_{i0} \cdot (1 - r_{id})^t, \tag{2}$$

where $a_{it}$ is the ESH for species $i$ after $t$ years, $a_{i0}$ is the area at time 0 and $r_{id}$ is the rate of deforestation for species $i$ per year and is calculated using the equation

$$r_{id} = \frac{p_{id}}{15}, \tag{3}$$

where $p_{id}$ is the proportion of a species ESH lost between 2000 and 2015.

For the 77 species affected by exploitation, we also added the expected population decline due to exploitation to the figure for habitat loss to derive the total expected population decline from both sources. We assumed zero impact of exploitation for non-exploited species. We classified all species with an expected population decline of >80% as Critically Endangered, 50% as Endangered, 30% as Vulnerable and 15% as Near Threatened, as per IUCN guidelines[27]. We also reclassified 4 species based on the extent of occupancy (EOO) we calculated from our 2015 forest extent (criterion B2), using a minimum convex polygon method, classifying any species with an EOO of less than 500 km² as endangered and less than 2000 km² as vulnerable. We take a more proactive approach and project future declines in these species based on the current developmental priorities of the Indonesian government[66].

Finally, we separated the species into regional endemics and species that occur more widely so we could make recommendations for changes to the Red List status only for regional endemics. Since our study does not assess population changes outside our study region, we calculated the proportion of the total species range that is in Sundaland and did not make Red List status recommendations for species with less than 80% of their range in our study area.

**Area protected**. To determine the amount of legal protection species are presently afforded in Sundaland, we calculated the ESH of each species for 2015 that fell within a protected area (PA) of IUCN category I–V based on data from the world database of protected areas[67]. As a conservative reanalysis, we then removed from the ESH within the PAs all areas that were within 5 km of the forest edge to account for the impacts of exploitation. We thus estimated PA coverage beyond 5 km of a forest edge.

**Species-area relationship analysis**. In line with previous analyses (e.g. Wilcove et al. and Brook et al.[12,68]) we used a species area relationship to estimate the number of extinctions expected by 2100 as result of habitat loss if it were to continue at the current rates.

$$S = cA^z, \tag{4}$$

where $S$ is the ratio of original to current species, $A$ is the ratio of original to current habitat and $c$ and $z$ are constants. We used a range of $z$-values for our calculation, given uncertainty regarding slope of the species area relationship[37] and the debate surrounding biodiversity responses to reducing area[69]. We used a lower value of 0.18 taken from[70] and an upper value of 0.35 which was used in ref. [12]; this range

also encompassed the value suggested by[37] of 0.21. We then applied these values to projected forest loss by 2100 for the whole regions and Borneo, Sumatra, Java, Bali and peninsular Malaysia individually.

**Code availability**. R and Python code for the analyses is available from the corresponding author upon request.

## Data availability
The authors declare that the data supporting the findings of this study are available within the paper and its supplementary information files. The updated species range maps used are available on request.

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

## Acknowledgements

We thank BirdLife International for curating and providing the range maps for this analysis. W.S.S. and L.R.C. were supported by a Ministry of Education of Singapore Tier 2 grant [MOE2015-T2–2–121].

## Author contributions

All (W.S.S., D.P.E., F.E.R., J.M., L.R.C.) authors developed the research questions, designed the analysis and interpreted results. F.E.R. and D.P.E. provided the expert information regarding species ecology and persecution. J.M. provided specialist mapping and remote sensing input. W.S.S. performed all coding and analysis and wrote the first draft of the manuscript, with all authors contributing substantially to manuscript revisions.

## Additional information

**Competing interests:** The authors declare no competing interests.

