## [Peer Review File · Nature Communications]

Reviewers' comments:

Reviewer #1 (Remarks to the Author):

Trapping for the pet trade is a major, although long underestimated, threat to biodiversity. This is the first analysis to combine the estimated effects of trapping and deforestation on animals at a regional scale. Most Red List assessments for tropical species are based on measurements of deforestation; the Red List does not do a good job of incorporating overexploitation in assessments in a quantitative way. The authors have come up with a workable method for doing this. I found three of the paper's results to be particularly interesting: (1) suggested changes in IUCN Red List status for many species, (2) an average of 83% of traded species' ranges are within 5 km of an edge, meaning they are probably subject to significant trapping pressure, and (3) trapping was a more serious threat than deforestation for 74% of commercially important lowland species. The paper has significant implications for assessment of extinction risk across the tropics.

But I do have some concerns.

1. I am not convinced that what the authors are calling "area of occupancy" is in accordance with the IUCN's definitions (<http://www.iucnredlist.org/technical-documents/categories-and-criteria/2001-categories-criteria>). It seems to be in between extent of occurrence and area of occupancy and closer to "extent of suitable habitat" (e.g. Beresford et al. 2011, Poor overlap between the distribution of Protected Areas and globally threatened birds in Africa, Animal Conservation). It is critical to get this right when using the Red List's thresholds for extinction risk assessment. As is, it appears that the authors are overestimating extinction risk by using a metric that is larger than AOO. Please review and cite the relevant literature (e.g. Gaton and Fuller 2008 J. Applied Ecol. The sizes of species' geographic ranges; Joppa et al. 2016 Conservation Biology, Impact of alternative metrics on estimates of extent of occurrence for extinction risk assessment) and clarify.

2. Also related to the Red List results, I imagine that people at the IUCN may not be eager to take on the suggested new classifications as they are presented. The authors wrote a few caveats about their Red List category results (e.g. for Straw-headed Bulbul, with which I agree), but it is not entirely clear how literally we should take the IUCN category suggestions. Which species need revised categories? Can this be specified in the supplementary tables? Also see my comment about the application of the IUCN criteria on lines 374-376.

3. And along these lines, how will the revised assessments actually be done? I could see the models fitting into criterion E (which is rarely used) or into the more typical criteria. It would be good if the authors discussed this so that their results have a better chance of being applied.

4. Throughout the text, figures, and choices of citations, trapping and hunting are often treated as synonymous. Trapping and hunting are fundamentally different processes (e.g. trapping is often non-lethal). I have highlighted some but not all of the occurrences of this in the manuscript. This needs to be clarified throughout.

Please see my specific comments on the manuscript file.

Bert Harris

[Editorial Note: Due to the journal policy, we are unable to publish annotated manuscript files as part of the Peer Review File.]

Reviewer #2 (Remarks to the Author):

See full review in the attachment.

[Editorial Note: Please see the Reviewer #2 comments on the following page.]

Review Symes et al. Nat. Com.

This is an interesting paper where the authors quantify the combined effects of deforestation and hunting/trapping on bird populations in the Sundaland region. I enjoyed reading the paper and I praise the authors for undertaking the substantial effort of assessing simultaneously both threats instead of looking at each in isolation. Undoubtedly, and besides other local efforts (Constantino 2016), here is where the novelty of this paper resides. This said, I have concerns with a number of aspects in the methodological approach, which I point out below.

- Framework of the paper. I think the authors should make more explicit that they are looking at the effects of trapping for cagebird trade, which is only made unambiguously clear in line 175. The introduction is full of background information about wildlife hunting in general, and towards the end of the intro, in line 75, they argue that deforestation and hunting impacts have been studied in isolation, but that the combined effects of wildlife trade and deforestation on extinction threats are likely underestimated. They should make a better transition between the previous articles they report on hunting impacts on wildlife in general, and on birds in particular and the specific case of trapping for wildlife trade, as bird hunting can be motivated for food consumption (Begazo and Bodmer 1998) but also for the use of feathers and casques for rituals and for the traditional medicine (Bennett et al. 1997; Buij et al. 2016)).
- Trapping impacts. The authors use a 5km threshold of hunting pressure on bird species based on Harris et al. 2016. This is a licit choice for the study region, where few remote areas are left and probably hunting distances are shorter than those reported in Benítez-López et al. 2017 (7 km). Although one could argue whether impacts may extend even longer, I am not going to ask the authors to redo their analyses to accommodate a threshold distance (7km) that is mostly based on many studies outside the region. However I am bit uncomfortable with the approach used to estimate the accessibility of the bird ranges to trappers. The authors indicate that this was calculated from the closest forest edge, and while this could be a valid assumption, I am more than certain that forest edges associated to roads (or settlements) are probably a better proxy for accessibility than all forest edges. I would like thus to ask the authors to either refine their analyses according to this comment or at least discuss what the implications of their choice are (overestimation of hunting impacts?).
- Trapping efficacy. Trapping pressure or efficacy should be correctly defined and units given. Is it expressed as the expected proportion of individuals removed? Additionally, this variable is based on the expert assessment of the commercial value of the bird species. However very little information is given on how and why these three categories were assigned. At least I would expect a table with the species name, the category, the source and the criteria used for assigning the commercial category. Also, it is not clear how the trapping pressure values (min and max) are assigned to each of the categories. I think the authors should clarify this in the Sup. Mat. Finally, I suggest the authors to state explicitly that trapping impacts are quantified only for the 77 commercially valuable species, otherwise I don't understand how the combined declines from deforestation and trapping are lower (ca. 25%, line 80) than the effect of trapping alone (37%, Line 75).
- IUCN red list assessment. I am particularly not convinced with the authors' approach to estimate population declines due to habitat loss. The authors assume that the "rate of habitat loss was directly proportional to the rate of population decline, such that a loss of 1% suitable habitat per year was equivalent to a 1% population decline per year." There are a number of studies that show that habitat loss is not directly translated into population decline, actually the relationship between both may be non-linear or threshold-dependent (Fahrig 2001; Fahrig 2002 or see Swift et al. 2009 for a review), where increasing habitat loss might not lead to a reduction in population size up to a certain threshold, at which a small decrease in habitat may result in a collapse in population densities. Actually, the response of species to habitat loss (and other environmental changes) is typically not instantaneous, usually populations decline gradually (and eventually go extinct) but not at the same rate as habitat decreases, and it might take some decades before a local population (or a species, in case of restricted distributions) disappear after the disturbance took place ("extinction debt" concept, Tilman et al. 1994; Hanski and Ovaskainen 2002). This brings quite some uncertainty on the results obtained by the authors and it is major drawback

in their analyses. Right now I cannot think of a clever way to solve this issue but I would encourage the authors to rethink on a better approach to account for population declines due to habitat loss.

- Rate of deforestation. This should be better defined. I thought you were looking at change in forest loss from 2000 to 2015, so how is this converted into a rate? Because you use this information to infer the rate in a period longer than 15 years, right? (Lines 362-364). Please, explain better.
- Area protected. The authors estimate the amount of protected area overlapping with the AOO of each species. For this analysis they use all protected areas with IUCN category I-VI, however protected areas under category VI are in many cases indigenous reserves where hunting is allowed. So probably the authors should remove this from their analyses.
- I found strange that the authors present and discuss an extra analysis in the discussion section to explore the number of species that will likely go extinct due to habitat loss (using species-area relationships). I think the analysis is a nice contribution to the article, so I would like to find it clearly reported in the methods and results section, together with the rationale for its inclusion.

Minor comments

Line 29 and 31: Shouldn't it be "tropical" biodiversity instead of "global"?

Line 75: please add "commercially valuable" before "species".

Lines 87-88: I couldn't agree more. In the light of these results, could we safely conclude that hunting/overexploitation is a more serious threat for wildlife populations (at least birds) than land use change/habitat loss?

Lines 213-215: This is an odd sentence, please rewrite.

Lines 219-220: I think this statement requires a reference.

Lines 231-232: this is in my opinion the main flaw of the paper. Actually I am not sure whether this assumption leads to conservative results, because probably population numbers either do not respond to small declines in habitat availability, or they do it in a time-lagged manner.

Lines 234: Indeed, they are probably synergistic. You could cite Brooks et al 2008 here.

Line 247: you did not look either to fragmentation impacts, e.g.: increased isolation of fragments and potential loss of functional connectivity. I see here a lot of potential for further research.

Line 342: what are PERT distributions? Please define.

Lines 344-346: Please add a reference to Sup. Table 1 and 2.

Line 386: Please add "We thus estimated PA coverage for the area beyond 5 km from an edge (to account for the effects of trapping)." Or similar.

Figures. All figures are clear and convey the main messages of the article. I appreciate that the maps are both attractive but also self-explanatory. I'd probably spend a bit of time on improving the quality of Figure 3.

Figure 3: Why do the lines extend beyond 5km? I thought that hunting impacts were quantified up to this threshold distance (lines 346-348).

Besides my criticism, I would like to reiterate that I enjoyed reading the article and I commend the authors for such a nicely written and conservation-relevant piece of work. We urgently need this kind of studies where several threats are accounted simultaneously. I will be happy to clarify or discuss my comments if necessary.

Best regards,

Ana Benítez

References

- Bennett, E. L., Nyaoi, A. J., & Sompud, J. (1997). Hornbills *Buceros* spp. and culture in northern Borneo: Can they continue to co-exist?. *Biological Conservation*, 82(1), 41-46.
- Begazo, A. J., & Bodmer, R. E. (1998). Use and conservation of cracidae (Aves: Galliformes) in the Peruvian Amazon. *Oryx*, 32(4), 301-309.
- Brook, B. W., Sodhi, N. S., & Bradshaw, C. J. (2008). Synergies among extinction drivers under global change. *Trends in ecology & evolution*, 23(8), 453-460.
- Constantino, P. (2016). Deforestation and hunting effects on wildlife across Amazonian indigenous lands. *Ecology and Society*, 21(2).
- Fahrig, L. (2001). How much habitat is enough?. *Biological conservation*, 100(1), 65-74.
- Fahrig, L. (2002). Effect of habitat fragmentation on the extinction threshold: a synthesis. *Ecological applications*, 12(2), 346-353.
- Hanski, I., & Ovaskainen, O. (2002). Extinction debt at extinction threshold. *Conservation biology*, 16(3), 666-673.
- Tilman, D., May, R. M., Lehman, C. L., & Nowak, M. A. (1994). Habitat destruction and the extinction debt. *Nature*, 371(6492), 65-66.

Reviewer #3 (Remarks to the Author):

In their manuscript, Symes et al. integrate information on forest area loss (i.e. habitat loss) with trapping pressure modelling to assess impacts on populations of bird species in the Sundaland biodiversity hotspot. The authors found that 89% of the 308 species studied experienced loss in forest habitat area. These 308 species were selected based on habitat preference (forest dependent) and elevational requirements (lowland species). The authors further estimated trapping effects on 77 of these species (i.e. the commercially valuable species). They subsequently estimated combined impacts of habitat loss and trapping on these 77 species but also across all other species. By linking the expected population declines as a consequence of both pressures to IUCN Red List criteria, the authors suggest that extinction risks of traded species may be seriously underestimated.

The study is providing an interesting and valuable conceptual approach to understanding likely consequences of area loss and hunting for biodiversity. The analyses are implemented in a region of high conservation importance. And the outputs are important not only for how we understand and create IUCN Red Lists but also for emphasizing the severe risks biodiversity continues to experience not only from land use change but also from human populations directly in regions that are highly diverse.

Yet, I find several aspects of the analyses puzzling, and despite reading across these sections several times, fail to follow the logic of some of the methods. One example is the restriction of parts of the analyses (effects of trapping) to subsets of the study region. Furthermore, I don't understand from the method description how the trapping pressure is computed for 77 species but trapping effects are subsequently rolled out across all of the 308 species. Perhaps I am missing a logic step here?

The findings of the majority of species being within 5 km of the forest edge (bearing in mind the resolution of the land cover data sets used) is intriguing and an important result. However, I am confused about the trapping effects (see detailed comment 6 below). I assume that the effects of hunting on species was only quantified for the Indonesian part of a species' range? Perhaps the authors need to be much clearer about these important details in the main text. So the authors actually only computed combined effects of habitat loss and trapping in Indonesia? And further, how many species do not have some part of their range in Indonesia and are hence excluded from that analysis? Some information on this has found its way into the main text, but it really is not easy to understand without studying the methods in detail (Lines 111-113). I guess this inconsistency in applying the method (assessing combined effects of trapping and area loss) across the study region begs the question, why the authors did not restrict themselves to the Indonesian region or alternatively, why did they not choose to do the combined analyses across the entire region and then compare results to the more conservative scenario of only allowing trapping effects to affect population size in Indonesia?

The forest-dependency of species analysed in this study determines the importance of key findings. It is therefore crucial to understand the labelling of forests as forest-dependent or forest specialists. However, the methods are not detailed enough to allow that understanding (see detailed comment 2). Figure 1 provides a nice and clear overview on the changes in area of occupancy presumably because of changes in habitat between 2000 and 2015. It would be good to see species highlighted that did not lose or alternatively increased their AOO across the IUCN categories. Some species did not experience habitat loss and trapping impacts (indicated as '0' in the Supp table), but would you expect them to increase in population size provided their habitat as extended? And who are these species (are they listed as threatened? would they become delisted based on the results?).

I am not sure whether I should find the statement on protected areas and their effectiveness in

conserving forests too vague or too ignorant of the wide literature on protected area effectiveness. We know that protected area effectiveness varies widely within and between geographic regions. Furthermore, the authors have data on forest habitats and their changes across the region, so could easily have put numbers on the effectiveness of protected areas in their study region?

A minor comment also refers to the title: does this really reflect the content of the study, as I would have thought hunting/trapping is done for a varied set of reasons including hunting for food?

Overall, I think the study is important and highlights the issue of trapping pressure affecting populations of wildlife in addition to effects resulting from area loss and fragmentation of habitats. However, I would need to see more detailed descriptions in the Methods.

Fig 2: the historic range ('purple line') - is that the range based on forest cover in 2000?

(1) How do the authors define the 'majority of the range' (is it 51%? 70%? XX?).

(2) The description of the 'forest-dependency assessments' leaves me with several questions: e.g. who are the experts who made those assessments? How many species were affected by being wrongly labelled according to those experts? And how do BirdLife International's assessment of forest dependency match the IUCN habitat classifications for the selected species? Answering those questions would help to get some information on uncertainty involved in the study. Is the final bird species list (with accompanying data) available for scrutiny by other bird experts?

(3) The forest cover change was computed from land cover maps with a spatial resolution of 250 m. The authors lumped all forest types to extract forest cover for 2000 and 2015. How good are these change maps for capturing forest degradation effects relevant for hunting pressure ('access to') compared to say 30 m resolution maps based on Landsat data?

(4) The authors used a DEM at 1 km resolution to refine ranges for the selected birds. How many birds had known elevation ranges and thus were included in this refinement? And why did the authors use this poor resolution dataset given the highly heterogeneous nature of landscape topography in some of the regions covered by the study? More clarification on this aspect would be helpful as elevation is also used to define the path distance raster and thus assess the proportion of a species' range, in which trapping could take place. See also Figure 2.

(5) The path distance raster: Perhaps this is not relevant for this analysis, but wouldn't you expect roads and potentially rivers to allow further access to forests independent of the location of the forest edge. With a 250 m resolution for the forests, any edge resulting from such roads would disappear, so it is unlikely (or I misunderstand) that the road / river effects would be captured from forest extent maps alone. But perhaps I misunderstand and roads/river allowing access to hunting are not important?

(6) The 77 species identified as important for trade: were market prices available for all of them? Can the authors provide more information on the parameter used in the PERT distribution modelling? I am also not sure why the authors assessed trapping as an issue in Indonesia only? I can understand that the assumptions made (e.g. 5 km distance, market prices) might only be available for Indonesia (although this is not clear from the Methods. But the analyses are implemented for other countries as well, in which hunting does occur. So by only using the method for one region, how does that allow you to compare across regions? And if the authors deem that approach too risky for countries outside Indonesia, why not focus their study on Indonesia only?

(7) The assumption of 1% population decline with 1% forest area decline across the range: is that a reasonable assumption given the many factors at play at this scale? Or perhaps the authors have evidence for that assumption?

Further Detailed Comments:

Line 29 Remote 'drives'

Line 47/48 I don't understand the logic behind the inclusion of this statement

Line 68: in the methods you imply that trapping as a factor for population declines was only estimated for Indonesia, but this is not obvious here. Furthermore, the statement might be confusing: so if trapping typically extends beyond 5 km, why only go up to 5 km?

Line 77: is that all 77 species?

Lines 82 ff: This is just wondering aloud: the species you classified as commercially valuable - were these all from Indonesia? And the areas in which you included trapping as a factor: was that only Indonesia? So if you find strong population declines, is this because your estimates are biased towards Indonesia?

Line 360 Please explain that criterion

Line 374 As someone not incredibly familiar with the intricacies of the Red List criteria and assessments, could the authors please provide more information on this section. Are the species reclassified from any of the classes that they were grouped into following the population decline assessment? How many species were reclassified? Which ones?

Lines 385/386: did you focus on Indonesia only for this (similar to your approach above?) What data did you use to delineate protected areas.

Please feel free to ask me for more details on some of my comments if they are unclear.

Marion Pfeifer

Reviewers' comments:

Reviewer #1 (Bert Harris, self-identified in review):

Trapping for the pet trade is a major, although long underestimated, threat to biodiversity. This is the first analysis to combine the estimated effects of trapping and deforestation on animals at a regional scale. Most Red List assessments for tropical species are based on measurements of deforestation; the Red List does not do a good job of incorporating overexploitation in assessments in a quantitative way. The authors have come up with a workable method for doing this. I found three of the paper's results to be particularly interesting: (1) suggested changes in IUCN Red List status for many species, (2) an average of 83% of traded species' ranges are within 5 km of an edge, meaning they are probably subject to significant trapping pressure, and (3) trapping was a more serious threat than deforestation for 74% of commercially important lowland species. The paper has significant implications for assessment of extinction risk across the tropics.

We are delighted that the Reviewer considers this a novel method and analysis with important implications for conservation assessments.

But I do have some concerns.

1. I am not convinced that what the authors are calling "area of occupancy" is in accordance with the IUCN's definitions (<http://www.iucnredlist.org/technical-documents/categories-and-criteria/2001-categories-criteria>). It seems to be in between extent of occurrence and area of occupancy and closer to "extent of suitable habitat" (e.g. Beresford et al. 2011, Poor overlap between the distribution of Protected Areas and globally threatened birds in Africa, Animal Conservation). It is critical to get this right when using the Red List's thresholds for extinction risk assessment. As is, it appears that the authors are overestimating extinction risk by using a metric that is larger than AOO. Please review and cite the relevant literature (e.g. Gaton and Fuller 2008 J. Applied Ecol. The sizes of species' geographic ranges; Joppa et al. 2016 Conservation Biology, Impact of alternative metrics on estimates of extent of occurrence for extinction risk assessment) and clarify.

Thank you for pointing out this inconsistency in our manuscript. The Reviewer is correct that we have effectively calculated the extent of suitable habitat rather than the area of occupancy as defined by the IUCN red list guidelines. To resolve this issue we have changed how we refer to our calculated area to "extent of suitable habitat" (ESH) to eliminate any possible confusion with the Red List defined AOO. Since our Red List status recommendations in all but four species rely on a calculation of decline in ESH, which is calculated consistently throughout our study, we are confident that we are not overestimating risk for the majority of our species. For the four species whose Red List reassessment was based on EOO in our analysis due to restricted range (the Enggano endemics), we have recalculated the EOO using the IUCN recommended method of minimum convex polygon calculation. We also recalculated the EOO for the two species of Mentawai endemic birds for which we were not able to calculate a change in extent of suitable habitat.

2. Also related to the Red List results, I imagine that people at the IUCN may not be eager to take on the suggested new classifications as they are presented. The authors wrote a few caveats about their Red List category results (e.g. for Straw-headed Bulbul, with which I agree), but it is not entirely clear how literally we should take the IUCN category suggestions. Which species need revised categories? Can this be specified in the supplementary tables? Also see my comment about the application of the IUCN criteria on lines 374-376.

We appreciate the opportunity to explain our results further. Some of the changes we have calculated are affected by certain (often species specific) caveats (Straw-headed Bulbul, Javan Blue-banded Kingfisher). To address this uncertainty, we have discussed some of our results in more detail in the main text (lines 118-123 Results, Lines 254-262 discussion). We have also included a new column in the supplementary tables for each status change, indicating our confidence in the status recommendation, and another with more detailed comments specific to each species.

*Lines 118-123: Most notably, these include Javan Blue-banded Kingfisher (*Alcedo euryzona*) from CR to LC, Silvery Pigeon (*Columba argentina*) from CR to EN, White-rumped Woodpecker (*Meiglyptes tristis*) from EN to LC, and Straw-headed Bulbul (*Pycnonotus zeylanicus*) from EN to VU. However, for some of these species, causes of endangerment are not covered appropriately by our methodology so that we do not recommend their downlisting based on our results (see Discussion).*

*Lines 254-262: In predicting losses to deforestation, the underlying maps, while the best available, likely contain commission and omission errors (e.g., areas that are either included or excluded erroneously), which could lead to inaccurate decline or exploitation estimates. Also, inaccuracies may occur where species' habitat requirements are not fully understood (e.g., Bonaparte's Nightjar (*Caprimulgus concretus*)). Our method is not sensitive to species that have very specialist habitat requirements, or exist at very low densities within suitable habitat. For example, the Javan Blue-banded Kingfisher (*Alcedo euryzona*), which is limited to lowland and hill riverine forest that is not specifically defined on our maps, is suggested to be downgraded from CR to LC in our analysis but riverine forest has been particularly affected by deforestation in the region. Consequently, while our results represent an improvement on existing knowledge in the majority of cases, some assessments must still be judged in a separate context, as would occur in any normal IUCN assessment process.*

3. And along these lines, how will the revised assessments actually be done? I could see the models fitting into criterion E (which is rarely used) or into the more typical criteria. It would be good if the authors discussed this so that their results have a better chance of being applied.

We agree with the Reviewer that additional information about the criteria is important, so we have added a column in supplementary table 1 to reflect this. We have also added a discussion of this in lines 296-302:

lines 296-302: In this study, we estimated the rates of population decline over 3 generations or 10 years, for assessment under the IUCN criterion A3, using an index of abundance and actual levels of persecution. The mode of assessment applied can make a significant difference to the end result. In our case, criterion E (a quantitative assessment of extinction risk) could be used via a species-area relationship (SAR) calculation, but we chose not to use this method because the SAR has been previously criticised for over-estimating extinction risk from habitat loss³⁸. However, reassessments using this criterion and more complicated extinction-risk models are a valuable area for future research.

4. Throughout the text, figures, and choices of citations, trapping and hunting are often treated as synonymous. Trapping and hunting are fundamentally different processes (e.g. trapping is often non-lethal). I have highlighted some but not all of the occurrences of this in the manuscript. This needs to be clarified throughout.

Apologies. It was not our intention to blur the lines between hunting and trapping, and we have revised the text to make the differences clearer. For the purposes of this analysis, we have assumed

that both hunting and trapping lead to a population decline (even though trapping for cagebirds is non lethal, in the short-term at least). We appreciate the terminology we had used was confusing, so we have replaced the word trapping with exploitation throughout the manuscript (where appropriate) and have added an explanation of what types of exploitation we are referring to in the methods (lines 396-406)

*Lines 396-406: To assess the impact of wildlife trade on commercially valuable species, we estimated the accessibility of the bird range to trappers and hunters. Hunting and trapping are fundamentally different processes, with different drivers and often different actors. In Sundaland, we categorised 3 types of threat: international trade, domestic trade and local hunting. We classify the 'domestic market' as trade within Indonesia, which is particularly important for songbirds and is dominated by cagebird trapping. International trade is important for a small number of commercially valuable species [e.g., Helmeted Hornbill *Rhinoplax vigil*]. Our final category, 'local hunting', is mostly perpetrated by small-scale actors, often opportunistically killing large-bodied species for consumption and/or selling on local markets. For a full breakdown of which species are in which category see supplementary table 1. We define exploitation in our study as a combination of all three of these processes since they all result in population declines.*

Please see my specific comments on the manuscript file.

We thank the Reviewer for his specific comments on the manuscript and have made the vast majority of the changes suggested (some were no longer needed due to other revisions).

To address the comment at the start of the Methods regarding repeatability of our study "How can you ensure that your study is repeatable? Will you archive a shapefile of your study area?", we would be happy to archive copies of the shapefiles we used. However, the data are currently owned and managed by Birdlife International and we currently do not have an agreement with them allowing us to do that. We will, however, investigate further as we agree that we should make our maps more widely available.

Reviewer #2 (Ana Benítez, self-identified in review):

This is an interesting paper where the authors quantify the combined effects of deforestation and hunting/trapping on bird populations in the Sundaland region. I enjoyed reading the paper and I praise the authors for undertaking the substantial effort of assessing simultaneously both threats instead of looking at each in isolation. Undoubtedly, and besides other local efforts (Constantino 2016), here is where the novelty of this paper resides. This said, I have concerns with a number of aspects in the methodological approach, which I point out below.

Thank you, we are very pleased the reviewer enjoyed our manuscript and appreciate the recognition of our substantial endeavours to simultaneously assess the threats. We are also happy the reviewer found our study to be novel, and we hope we can address her concerns below.

1. - Framework of the paper. I think the authors should make more explicit that they are looking at the effects of trapping for cagebird trade, which is only made unambiguously clear in line 175. The introduction is full of background information about wildlife hunting in general, and towards the end of the intro, in line 75, they argue that deforestation and hunting impacts have been studied in isolation, but that the combined effects of wildlife trade and deforestation on extinction threats are likely underestimated. They should make a better transition between the previous articles they

report on hunting impacts on wildlife in general, and on birds in particular and the specific case of trapping for wildlife trade, as bird hunting can be motivated for food consumption (Begazo and Bodmer 1998) but also for the use of feathers and casques for rituals and for the traditional medicine (Bennett et al. 1997; Buij et al. 2016)).

We are not explicitly only measuring the impact of trapping for the cage bird trade, instead we are trying to assess the impact of multiple sources of hunting and trapping threat. However, we completely agree with the Reviewer that greater clarity regarding this point is needed in our manuscript at a much earlier stage, and that we can improve on the discussion regarding the difference between the varying sources of hunting and trapping threat. While trapping for the cage bird trade is the largest source of threat in the region, hunting of other large species for food and/or export (especially hornbills) is also a large source of threat for some species. We have tried to make this clearer in the revised manuscript by changing the way we refer to the losses from multiple sources of exploitation in the Introduction lines (36-39, 58-60), by adding information on these sources to the supplementary tables, and by adding a section in the methods (lines 396-406) making it explicit which sources of declines from hunting and trapping are important for which species.

Lines 36-39: Illegal hunting of wildlife for internationally traded products, pets and food is directly responsible for the declines of emblematic species such as elephant¹⁰, rhinoceros¹¹, tiger¹², and bali starling¹³

Lines 58-60: Focusing on 308 forest dependent bird species, 77 of which are heavily trapped for the cagebird trade, animal products or as a local food resource, we quantitatively assess the compounded effect of habitat loss and exploitation on their extinction risk.

*Lines 396-406: To assess the impact of wildlife trade on commercially valuable species, we estimated the accessibility of the bird range to trappers and hunters. Hunting and trapping are fundamentally different processes, with different drivers and often different actors. In Sundaland, we categorised 3 types of threat: international trade, domestic trade and local hunting. We classify the 'domestic market' as trade within Indonesia, which is particularly important for songbirds and is dominated by cagebird trapping. International trade is important for a small number of commercially valuable species [e.g., Helmeted Hornbill *Rhinoplax vigil*]. Our final category, 'local hunting', is mostly perpetrated by small-scale actors, often opportunistically killing large-bodied species for consumption and/or selling on local markets. For a full breakdown of which species are in which category see supplementary table 1. We define exploitation in our study as a combination of all three of these processes since they all result in population declines.*

2. - Trapping impacts. The authors use a 5km threshold of hunting pressure on bird species based on Harris et al. 2016. This is a licit choice for the study region, where few remote areas are left and probably hunting distances are shorter than those reported in Benítez-López et al. 2017 (7 km). Although one could argue whether impacts may extend even longer, I am not going to ask the authors to redo their analyses to accommodate a threshold distance (7km) that is mostly based on many studies outside the region. However I am bit uncomfortable with the approach used to estimate the accessibility of the bird ranges to trappers. The authors indicate that this was calculated from the closest forest edge, and while this could be a valid assumption, I am more than certain that forest edges associated to roads (or settlements) are probably a better proxy for accessibility than all forest edges. I would like thus to ask the authors to either refine their analyses according to this comment or at least discuss what the implications of their choice are (overestimation of hunting impacts?).

We used the 5 km threshold for two reasons: firstly, because that was what the empirical evidence suggests is local hunting behaviour (eg. Harris et al 2017); and secondly because we wanted our results to be on the conservative side, given the issues with our methodology highlighted by Reviewer 2. We are pleased the reviewer agrees with our 5km threshold. We also agree that it is likely hunters and trapper will travel further from the forest edge, especially to trap high value species (as was highlighted for the Barusan shama in Eaton et al. 2016).

The reviewer is, of course, correct that realistically access is defined by a complex interaction of many different factors including roads, rivers and towns. However, several issues make calculating a more complicated metric unfeasible. Good maps of roads for the region do not exist, with many important access points missing (see Ibisch et al. 2016). As such, decline estimates based on these maps would likely be inaccurate.

In the revised manuscript, we have additionally calculated access based on various hybrid maps of the region, which combine data from open street map and the world resources institute. The results of these analyses are presented in the supplementary information. We feel that these results highlight the key inaccuracies of currently available maps and justify our original methodological choice to use forest edges. We have also added lines 225-244, which discuss this issue in more detail.

Lines 225-244: While our method is a rapid and straightforward way for assessing population declines, there are several limitations that must be noted. In assessing the impact of exploitation, detailed information on the behaviour of wild bird trappers and species responses to exploitation is not available. Hence, we made simplified assumptions regarding hunting impact and accessibility, but in reality, species responses to hunting are more nuanced than the three categories we used (low, medium and high)⁴² and accessibility is a complex interaction of population, roads, topography and markets. We attempted to account for this uncertainty by combining maps of major roads and all available roads to calculate additional metrics of accessibility.

Using road maps instead of forest edges made a considerable difference to our results, with both road maps leading to considerably lower threat estimates. When exclusively using major roads, only 16 species were above Red List thresholds (6 EN and 10VU); when using all roads, this number rose to 38 (3 CR, 13 EN, 22VU) (see supplementary figures 4 and 5). However, the maps we used, while the best available (open street map and WRI produced Indonesia map), are fundamentally inaccurate, with many roads missing⁴³. Crucially, this inaccuracy is not uniform and changes the analysis in biased and unpredictable ways. For example, the maps contain no roads inside protected areas in Java, suggesting Javanese forests are much more isolated than they actually are, leading to large underestimations of threat. Secondly, deciding to what extent individual roads are accessible to hunters requires further assumptions. By using distance from forest edge we avoid using maps that are known to be inaccurate⁴³ and we can account for access from other structures (such as rivers) by assuming deforestation follows these points of access.

3. - Trapping efficacy. Trapping pressure or efficacy should be correctly defined and units given. Is it expressed as the expected proportion of individuals removed? Additionally, this variable is based on the expert assessment of the commercial value of the bird species. However very little information is given on how and why these three categories were assigned. At least I would expect a table with the species name, the category, the source and the criteria used for assigning the commercial category. Also, it is not clear how the trapping pressure values (min and max) are assigned to each of the

categories. I think the authors should clarify this in the Sup. Mat. Finally, I suggest the authors to state explicitly that trapping impacts are quantified only for the 77 commercially valuable species, otherwise I don't understand how the combined declines from deforestation and trapping are lower (ca. 25%, line 80) than the effect of trapping alone (37%, Line 75).

We thank the Reviewer for pointing out that some aspects of the results were confusing. We have attempted to clarify these sections and hopefully it is now clear that we only calculated the impact of exploitation for the 77 commercially valuable species. We have added information to the Supplementary Table, breaking down the source of exploitation for all the commercially valuable species in our study, and we have included explanations of these categories in the Methods (lines 396-406).

*Lines 396-406: To assess the impact of wildlife trade on commercially valuable species, we estimated the accessibility of the bird range to trappers and hunters. Hunting and trapping are fundamentally different processes, with different drivers and often different actors. In Sundaland, we categorised 3 types of threat: international trade, domestic trade and local hunting. We classify the 'domestic market' as trade within Indonesia, which is particularly important for songbirds and is dominated by cagebird trapping. International trade is important for a small number of commercially valuable species [e.g., Helmeted Hornbill *Rhinoplax vigil*]. Our final category, 'local hunting', is mostly perpetrated by small-scale actors, often opportunistically killing large-bodied species for consumption and/or selling on local markets. For a full breakdown of which species are in which category see supplementary table 1. We define exploitation in our study as a combination of all three of these processes since they all result in population declines.*

4. - IUCN red list assessment. I am particularly not convinced with the authors' approach to estimate population declines due to habitat loss. The authors assume that the "rate of habitat loss was directly proportional to the rate of population decline, such that a loss of 1% suitable habitat per year was equivalent to a 1% population decline per year." There are a number of studies that show that habitat loss is not directly translated into population decline, actually the relationship between both may be nonlinear or threshold-dependent (Fahrig 2001; Fahrig 2002 or see Swift et al. 2009 for a review), where increasing habitat loss might not lead to a reduction in population size up to a certain threshold, at which a small decrease in habitat may result in a collapse in population densities. Actually, the response of species to habitat loss (and other environmental changes) is typically not instantaneous, usually populations decline gradually (and eventually go extinct) but not at the same rate as habitat decreases, and it might take some decades before a local population (or a species, in case of restricted distributions) disappear after the disturbance took place ("extinction debt" concept, Tilman et al. 1994; Hanski and Ovaskainen 2002). This brings quite some uncertainty on the results obtained by the authors and it is major drawback in their analyses. Right now I cannot think of a clever way to solve this issue but I would encourage the authors to rethink on a better approach to account for population declines due to habitat loss.

The Reviewer raises a good point and highlights a challenge for our analysis. The non-linearity and unpredictability of population responses to habitat loss pose a challenge for conservation policy makers and practitioners. While critical thresholds for habitat loss exist, they are very difficult and data intensive to calculate, and, for this reason, while the concept is ecologically interesting, it is of questionable utility to conservation decisions (as Swift et al 2009 attest). As to the question of time lags in population responses, these again are not especially useful to conservation practitioners, as from a management perspective (except in the case of a restoration project), a committed decline after a certain time lag can be considered the same as a decline that has already happened (REF?).

For these reasons, we feel that the assumption of linearity in this case is justified as it is the translation of land cover change into declines that is most useful to conservation practitioners. Furthermore, the assumption of linearity is consistent with the application of the precautionary principle, a concept central to the Convention on Biological Diversity.

Secondly, we would like to thank the Reviewer for highlighting the issue of extinction debt. However we respectfully disagree with its implications for our results. Extinction debt refers to populations of species in remnant habitat patches that are already committed to extinction based on edge effects and inbreeding depression even though they remain in existence for a short additional time. By not considering this process, we have included many areas where populations are already committed to extinction, rendering our results more conservative.

On the other hand, we agree that the potential for differing species responses to habitat loss was not explained sufficiently in the manuscript and that our assumption of linearity needs to be justified. To address this issue, we have added a rigorous explanation (lines 272-287) to the Discussion. We are confident that these changes should satisfy the Reviewer's concerns.

Lines 272-287: We also assume a linear relationship between deforestation and population decline, which we believe is a conservative assumption given the negative impacts that edge, area and isolation effects have on species⁴⁷. There is considerable debate as to whether this relationship is linear, with many theoretical (and some field) studies suggesting that populations can remain reasonably stable until a certain threshold of habitat loss (for a comprehensive review see⁴⁸), leading to overestimates of population decline under linear assumptions. However, while habitat thresholds may exist for some species in our study, given our current knowledge, calculating these thresholds accurately is impossible and as such they cannot be incorporated and would be of questionable utility to conservation decisions⁴⁸. Instead, our assumption of linearity is in line with the precautionary principle (which is acknowledged in the Convention of Biological Diversity), since it ensures we are not underestimating declines by assuming a threshold that does not exist, and thus is more useful to conservation decisions. For simplicity, we also assumed that the effects of the loss of habitat from deforestation and exploitation were additive, which results in three species having maximum estimated declines above 100%. In reality, the proportion of the remaining area of a species subjected to exploitation will increase as the habitat is reduced and fragmented, meaning the impacts are likely synergistic⁴⁹, making our estimates for most species conservative.

5. - Rate of deforestation. This should be better defined. I thought you were looking at change in forest loss from 2000 to 2015, so how is this converted into a rate? Because you use this information to infer the rate in a period longer than 15 years, right? (Lines 362-364). Please, explain better.

We apologise for the confusion here and have clarified this issue in lines 454-456 with equation 3.

Lines 454-456: deforestation for species i per year and is calculated using the equation

$$r_{id} = \frac{p_{id}}{15} \quad (3)$$

where p_{id} is the proportion of a species ESH lost between 2000 and 2015.

6. - Area protected. The authors estimate the amount of protected area overlapping with the AOO of each species. For this analysis they use all protected areas with IUCN category I-VI, however protected areas under category VI are in many cases indigenous reserves where hunting is allowed. So probably the authors should remove this from their analyses.

This is a good point. We have recalculated this statistic. In doing so, this has strengthened the overall pattern of our results (i.e., hunting was still a bigger driver of extinction risk than habitat loss).

7. - I found strange that the authors present and discuss an extra analysis in the discussion section to explore the number of species that will likely go extinct due to habitat loss (using species-area relationships). I think the analysis is a nice contribution to the article, so I would like to find it clearly reported in the methods and results section, together with the rationale for its inclusion.

As suggested, we have moved this section from the SI to the main Methods (478-488) and Results(68-71) section.

Lines 60-71: Using a simple reverse species area relationship our analysis suggests between 16.9% (52 species) and 30.1% (92) of all forest-dependent species will go extinct in the region by 2100 (see Supplementary Figure 3).

Minor comments

Line 29 and 31: Shouldn't it be "tropical" biodiversity instead of "global"?

Changed

Line 75: please add "commercially valuable" before "species".

Changed

Lines 87-88: I couldn't agree more. In the light of these results, could we safely conclude that hunting/overexploitation is a more serious threat for wildlife populations (at least birds) than land use change/habitat loss?

We certainly think so. It thus underscores the importance of our approach to assessing this risk in combination with land-use change.

Lines 213-215: This is an odd sentence, please rewrite.

Rewritten as follows (now lines 225-226): While our method is a rapid and straightforward way for assessing population declines, there are several limitations that must be noted

Lines 219-220: I think this statement requires a reference.

added

Lines 231-232: this is in my opinion the main flaw of the paper. Actually I am not sure whether this assumption leads to conservative results, because probably population numbers either do not respond to small declines in habitat availability, or they do it in a time-lagged manner.

We have added a justification for this assumption, and it has been discussed above in response to point 4. We also feel that in all cases the changes in habitat were not small (the average decline was around 50,000km²), and thus it is highly unlikely there was no population response. ?

Lines 234: Indeed, they are probably synergistic. You could cite Brooks et al 2008 here.

Thank you for the suggestion; we have added the reference.

Line 247: you did not look either to fragmentation impacts, e.g.: increased isolation of fragments and potential loss of functional connectivity. I see here a lot of potential for further research.

We agree, there is a lot of potential for further research. We now make this clearer in our Discussion (Line 309-313): The same is also true for the impacts of increasing fragmentation, which we do not directly consider, but are likely to have profound implications for many species⁴⁷. Incorporating logging impacts and fragmentation effects represents another important frontier in combined species assessment, especially in logging ravaged regions in the tropics (i.e. Southeast Asia, Congo, Southern and Eastern Amazon).

Line 342: what are PERT distributions? Please define.

We have added a short definition thus (Line 419-422): We used these estimates to parameterize PERT (programme evaluation and review technique) distributions (shape = 4). PERT distributions assign low probabilities to extreme values and thus are useful for assessing the uncertainty in expert estimates.

Lines 344-346: Please add a reference to Sup. Table 1 and 2.

Added

Line 386: Please add “We thus estimated PA coverage for the area beyond 5 km from an edge (to account for the effects of trapping).” Or similar.

Added

Figures. All figures are clear and convey the main messages of the article. I appreciate that the maps are both attractive but also self-explanatory. I’d probably spend a bit of time on improving the quality of Figure 3.

We appreciate Figure 3 was the least attractive and have endeavoured to improve on its quality. We changed the underlying data so that rather than having a single point for hunting impact at each km it now shows the full range of values that make up hunting impact from 0-5km. The result is a smoother curve and we hope you agree that it is now of a higher quality.

Figure 3: Why do the lines extend beyond 5km? I thought that hunting impacts were quantified up to this threshold distance (lines 346-348).

This is because our method actually calculated the total distance from the edge for the entire range of each species. However, including the information all the way up until 10km is confusing given the 5km limit we have used. This figure has been redrawn.

Besides my criticism, I would like to reiterate that I enjoyed reading the article and I commend the authors for such a nicely written and conservation-relevant piece of work. We urgently need this kind of studies where several threats are accounted simultaneously. I will be happy to clarify or discuss my comments if necessary.

We appreciate this comment and the kind offer for clarification.

Reviewer #3 (Marion Pfeifer, self-identified in review):

In their manuscript, Symes et al. integrate information on forest area loss (i.e. habitat loss) with trapping pressure modelling to assess impacts on populations of bird species in the Sundaland

biodiversity hotspot. The authors found that 89% of the 308 species studied experienced loss in forest habitat area. These 308 species were selected based on habitat preference (forest dependent) and elevational requirements (lowland species). The authors further estimated trapping effects on 77 of these species (i.e. the commercially valuable species). They subsequently estimated combined impacts of habitat loss and trapping on these 77 species but also across all other species. By linking the expected population declines as a consequence of both pressures to IUCN Red List criteria, the authors suggest that extinction risks of traded species may be seriously underestimated.

The study is providing an interesting and valuable conceptual approach to understanding likely consequences of area loss and hunting for biodiversity. The analyses are implemented in a region of high conservation importance. And the outputs are important not only for how we understand and create IUCN Red Lists but also for emphasizing the severe risks biodiversity continues to experience not only from land use change but also from human populations directly in regions that are highly diverse.

We are pleased the reviewer found our manuscript to be interesting and valuable. We are also glad the reviewer considers our results to be important for how we understand and create the IUCN Red Lists and the multifaceted risks biodiversity faces from human populations.

1. Yet, I find several aspects of the analyses puzzling, and despite reading across these sections several times, fail to follow the logic of some of the methods. One example is the restriction of parts of the analyses (effects of trapping) to subsets of the study region. Furthermore, I don't understand from the method description how the trapping pressure is computed for 77 species but trapping effects are subsequently rolled out across all of the 308 species. Perhaps I am missing a logic step here?

We agree there was a lack of clarity around these sections of the Methods and we have clarified these two issues. In summary here, we have added a clarifying statement (lines 427-431) that – because of cultural reasons – trapping pressure is considerably higher in some South-east Asian countries than others, and we therefore adjusted our analysis of the effects of trapping by the proportion of the range inside those highly affected countries. The non-application of these analyses to other South-east Asian countries is a conservative approach, because some exploitation pressure will apply to those countries as well. Furthermore, we have added a clarification (lines 459) that for all species other than the 77 commercially valuable (exploited) species, the impact of trapping and hunting was set to zero in our analysis since these species are not targeted. We hope that this is now much clearer in the Results section.

Lines 427-431: Since exploitation effort differs across the countries in the region, with dramatically more trade in Indonesia than Malaysia^{64,65}, we conservatively assumed exploitation was only an issue in Indonesia and adjusted our estimates of exploitation by the proportion of the range inside Indonesia. We therefore calculated the population reduction due to exploitation using:

Line 459: We assumed zero impact of exploitation for non-exploited species.

2. The findings of the majority of species being within 5 km of the forest edge (bearing in mind the resolution of the land cover data sets used) is intriguing and an important result. However, I am confused about the trapping effects (see detailed comment 6 below). I assume that the effects of hunting on species was only quantified for the Indonesian part of a species' range? Perhaps the authors need to be much clearer about these important details in the main text. So the authors actually only computed combined effects of habitat loss and trapping in Indonesia?

The exploitation effects were calculated over the entire range of the species within the biogeographic region of Sundaland, regardless of country. We then adjusted the estimation based on the proportion of the extent that was inside Indonesia, to account for differing rates of hunting in the region. We have clarified this point of the Methods.

Line 407-417 We created a path distance raster (with cell size of 150 m) for each of the range maps. The value of each cell in a path distance raster is the distance from the middle of that cell to the closest edge of the forest taking into account changes in elevation. From this raster, we calculated the percentage of the entire range for the 77 exploited species that was within a given distance from the forest edge, which we used to assess the proportion of the bird species' range in which exploitation could take place.

3. And further, how many species do not have some part of their range in Indonesia and are hence excluded from that analysis?

There is only one species which is regionally endemic, exploited and does not occur in Indonesia: Malay Peacock-pheasant. We likely underestimate the threat to this one species, but this limitation is clearly discussed in lines 245-253 and highlighted in the new additional notes we added to supplementary table 1. We have also now highlighted this one example in the Discussion to account for the Reviewer's concern.

*lines 245-253: Because our knowledge of trade dynamics and trapper behaviour is best in Indonesia, we made the conservative assumption that there was no exploitation outside Indonesia, but this is certainly not the case for many species^{27,28,44} and – at worst – led to unrealistic estimates in globally endangered species that are either highly persecuted outside of Indonesia (e.g. Straw-headed Bulbul *Pycnonotus zeylanicus*²⁸) and/or have the majority of their range in Malaysia (e.g. Malay Peacock-pheasant, *Polyplectron malacense*). Despite these limitations, our results agree with a recent meta-analysis of hunting, which showed global average population declines of 58% (versus 36.6% in this study), and depletion within 7 km of access points, further highlighting the role of the pet trade in driving defaunation^{9,45}.*

4. Some information on this has found its way into the main text, but it really is not easy to understand without studying the methods in detail (Lines 111-113). I guess this inconsistency in applying the method (assessing combined effects of trapping and area loss) across the study region begs the question, why the authors did not restrict themselves to the Indonesian region or alternatively, why did they not choose to do the combined analyses across the entire region and then compare results to the more conservative scenario of only allowing trapping effects to affect population size in Indonesia?

We do not believe there is an inconsistency in the application of the Method. Perhaps this stems from the lack of clarity in our previous manuscript version as discussed above, so we hope that our action in response to points 1-3 has addressed this issue. Restricting ourselves to Indonesia would have severely hampered the usefulness of the study to conservation practitioners as the majority of species are not restricted to only one of the 4 countries in the biogeographic region. Any results would therefore be of more limited use to conservation policy (for example the Red List) if we had failed to account for all of Sundaland. We refer to points 1-3 (see above) for the extensive revision and additions we have implemented to improve on clarity.

5. The forest-dependency of species analysed in this study determines the importance of key findings. It is therefore crucial to understand the labelling of forests as forest-dependent or forest

specialists. However, the methods are not detailed enough to allow that understanding (see detailed comment 2).

We appreciate there is some lack of clarity over this. In summary we took the Birdlife forest dependency categories as our base and then further refined this to species that are entirely dependent on forest, based on the expert opinion of two local ornithologists with extensive field experience. We have clarified this in lines 347-359 of the Methods.

*lines 347-359: We obtained range maps and ecological data for all species of birds occurring in Sundaland from BirdLife International using the most recent 2017 data⁴⁶. We then filtered our list to include only lowland forest specialist species, defined as those species with the majority of their range below 500 m above sea level and those described as being forest dependent according to BirdLife International⁵⁹. This list was further refined based on expert opinion (F.E.R. and D.P.E.) to remove/add species with inaccurate forest dependency assessments. Our assessment led to the inclusion of four species which had been classified as non-forest by BirdLife: White-crowned Forktail (*Enicurus leschenaultia*), Lesser Fish-eagle (*Ichthyophaga humilis*), Grey-headed Fish-eagle (*Ichthyophaga icthyaetus*), and Grey-cheeked Green-pigeon (*Treron griseicauda*). We also excluded forest dependent species that can be found breeding in forest plantations. We adhered to the BirdLife taxonomic treatment to ensure our results and recommendations are policy relevant. After filtering of upland and non-forest species, 308 lowland forest specialist species remained for analysis.*

6. Figure 1 provides a nice and clear overview on the changes in area of occupancy presumable because of changes in habitat between 2000 and 2015. It would be good to see species highlighted that did not lose or alternatively increased their AOO across the IUCN categories. Some species did not experience habitat loss and trapping impacts (indicated as '0' in the Supp table), but would you expect them to increase in population size provided their habitat as extended? And who are these species (are they listed as threatened? would they become delisted based on the results?).

*Thank you for this useful suggestion. To clarify, the species in Figure 1 were chosen for a number of reasons, most importantly because they are all persecuted species, hence demonstrating that the range within 5 km is relevant information. This figure does not show the change due to habitat loss, it simply shows the difference between the Birdlife range and our extent of suitable habitat in 2015, and the proportion of that extent which is within 5km. One of the species in the figure, *Alophoixus bres*, does not lose ESH over the study period and there is in fact a small gain. However, there are two major caveats: firstly, the increase in ESH is almost certainly entirely due to the slightly different mapping procedures used between 2000 and 2015, and therefore likely does not represent an actual increase in ESH; and secondly, all of the species that experienced an 'increase' are either from Java, an island with already extremely limited forest extent, or are restricted to other small islands. Thus we do not expect any substantial population increases in these species. We have clarified this in the discussion (lines 256-262), and added more information on each species in the SI to provide additional details important to our Results. The zeros in the original tables were due to a coding error and we thank the Reviewer for drawing this to our attention.*

lines 256-262: The changes in mapping methodology outlined in the methods also led to conservative underestimates of ESH decline, at its most extreme 23 species (Javanese endemics and range restricted Island species) experienced an increase in ESH between 2000 and 2015. However, all the species affected are restricted to extremely limited areas which have likely remained largely

unchanged in the analysis period. We therefore do not expect any substantial population increases in these species. Instead the increases highlight how our estimated decreases in other species are likely conservative.

7. I am not sure whether I should find the statement on protected areas and their effectiveness in conserving forests too vague or too ignorant of the wide literature on protected area effectiveness. We know that protected area effectiveness varies widely within and between geographic regions. Furthermore, the authors have data on forest habitats and their changes across the region, so could easily have put numbers on the effectiveness of protected areas in their study region?

We feel the reviewer may have misunderstood the statement: we do not make comments regarding the ability of protected areas (PA) to prevent deforestation as evidence on this across the region is not exactly clear, and this is not something our results allow us to comment on (see below). Instead we are referring to the ability of PAs in the region to prevent exploitation, which most evidence and the sources we cite suggest they do not. We have increased the breadth of evidence for our statements and have tried to clarify the sentence to prevent misunderstanding:

Lines 201-203: Finally, a lack of funding (annual shortfall of US\$521 million per year in Indonesia) for patrols and insufficient law enforcement and punishment of exploitation means that many PAs do not effectively prevent trapping and hunting^{9,21,29,35}.

With respect to the second assertion regarding PA effectiveness, unfortunately, assessing the effectiveness of protected areas is not possible using our methodology. Effectiveness of PAs based on a simple calculation of deforestation is potentially misleading as it lacks a counterfactual basis and PAs are often located in areas with less deforestation pressure anyway (Joppa & Pfaff 2009). Given the extensive methodological discussion around how to measure PA effectiveness accurately (see (Andam et al. 2008; Geldmann et al. 2013; Brun et al. 2015; Pfaff et al. 2016)), we respectfully disagree that we could 'easily have put numbers on the effectiveness of protected areas in their study region' as the Reviewer suggests.

- Andam KS, Ferraro PJ, Pfaff A, Sanchez-Azofeifa GA, Robalino J a. 2008. Measuring the effectiveness of protected area networks in reducing deforestation. *Proceedings of the National Academy of Sciences of the United States of America* **105**:16089–16094.
- Brun C, Cook AR, Lee JSH, Wich S a., Koh LP, Carrasco LR. 2015. Analysis of deforestation and protected area effectiveness in Indonesia: A comparison of Bayesian spatial models. *Global Environmental Change* **31**:285–295. Elsevier Ltd. Available from <http://linkinghub.elsevier.com/retrieve/pii/S0959378015000230>.
- Geldmann J, Barnes M, Coad L, Craigie ID, Hockings M, Burgess ND. 2013. Effectiveness of terrestrial protected areas in reducing habitat loss and population declines.
- Joppa LN, Pfaff A. 2009. High and far: Biases in the location of protected areas. *PLoS ONE* **4**:1–6.
- Pfaff A, Santiago-Ávila F, Joppa L. 2016. Evolving Protected-Area Impacts in Mexico: Political Shifts as Suggested by Impact Evaluations. *Forests* **8**:17. Available from <http://www.mdpi.com/1999-4907/8/1/17>.

8. A minor comment also refers to the title: does this really reflect the content of the study, as I would have thought hunting/trapping is done for a varied set of reasons including hunting for food?

It is done for a variety of reasons, but in our study region the most important and largest driver of decline is for wildlife trade both domestic and international. Therefore we feel the title is an accurate reflection of the content.

9. Overall, I think the study is important and highlights the issue of trapping pressure affecting populations of wildlife in addition to effects resulting from area loss and fragmentation of habitats. However, I would need to see more detailed descriptions in the Methods.

We thank the reviewer for agreeing that this is an important study and hope that we have now provided sufficient methodological detail in our revision.

10. Fig 2: the historic range ('purple line') - is that the range based on forest cover in 2000?

The purple line represents the outline of the historic range as provided by BirdLife international, it is not based on forest cover, we agree the figure legend was perhaps unclear and has been subsequently clarified:

*Fig 2. The three panels illustrate the ranges accessible to trappers for three species in our analysis: a) Sumatran Leafbird (*Chloropsis media*), b) Melodious Bulbul (*Alophoixus bres*) and c) White-crowned Hornbill (*Berenicornis comatus*). The purple line is the outline of the species' historic range (as provided by BirdLife International). The green area, which is divided into two shades, indicates the total extent of suitable habitat for the species in 2015, once it has been refined for current forest extent and elevation. The dark green regions are areas that are further than 5 km from the forest edge and considered inaccessible to trappers; the light green areas are regions that are within 5 km of a forest edge where exploitation is likely taking place. Species illustrations are reproduced with permission from 32.*

Further comments

(1) How do the authors define the 'majority of the range' (is it 51%? 70%? XX?).

It is 80%, we have updated the figure legend to include this.

(2) The description of the 'forest-dependency assessments' leaves me with several questions: e.g. who are the experts who made those assessments? How many species were affected by being wrongly labelled according to those experts? And how do BirdLife International's assessment of forest dependency match the IUCN habitat classifications for the selected species? Answering those questions would help to get some information on uncertainty involved in the study. Is the final bird species list (with accompanying data) available for scrutiny by other bird experts?

The species list is available in the supplementary tables, for other experts to assess. The experts are the two co-authors, David Edwards and Frank Rheindt, and this has been clarified in line 350. On the whole, the Birdlife categorisations match well, with only four species where our experts disagreed with Birdlife's 'non-forest' classification (note that Birdlife and IUCN classifications are the same). A more complete list of the species is available in supplementary tables.

(3) The forest cover change was computed from land cover maps with a spatial resolution of 250 m. The authors lumped all forest types to extract forest cover for 2000 and 2015. How good are these change maps for capturing forest degradation effects relevant for hunting pressure ('access to') compared to say 30 m resolution maps based on Landsat data?

The main advantage of these maps is that they outline primary forest areas, which we focus on. There are currently no 30 m resolution regional level products available for the study region that outline primary forest and secondary forest areas in 2000 and 2015. The 30 m resolution maps typically classify tree cover. This is a very serious limitation of using these maps in our study, given the vast extent of tree plantations, agroforestry and home gardens with trees in the region. It is true that the maps used in this study do not provide information on the level of degradation of any given forest area. But this information is not available in any resolution on regional level. The 30 m resolution maps would certainly show more openings in the forest cover, but we believe that the 250 m resolution data used in the study provided the best compromise for our analysis by providing the outlines of primary and secondary forest areas, and presenting major edges and openings (e.g. major openings for roads or habitation) within forest areas. We have used what we believe to be the best available data.

(4) The authors used a DEM at 1 km resolution to refine ranges for the selected birds. How many birds had known elevation ranges and thus were included in this refinement? And why did the authors use this poor resolution dataset given the highly heterogeneous nature of landscape topography in some of the regions covered by the study? More clarification on this aspect would be helpful as elevation is also used to define the path distance raster and thus assess the proportion of a species' range, in which trapping could take place. See also Figure 2.

A total of 88 species did not have known elevational ranges, but the underlying maps of their extent of occurrence supplied by Birdlife already accounted for this in a broad sense. We used a 1km resolution for computational reasons. Our study area covers over 1.3million km² (130million ha), and the analysis required by our study is computationally intensive and therefore we had to make decisions regarding the trade-off between precision and feasibility. We have updated the analysis using an elevational resolution of 90m, and it now takes ~10x longer, any higher resolution analysis would be unfeasible.

(5) The path distance raster: Perhaps this is not relevant for this analysis, but wouldn't you expect roads and potentially rivers to allow further access to forests independent of the location of the forest edge. With a 250 m resolution for the forests, any edge resulting from such roads would disappear, so it is unlikely (or I misunderstand) that the road / river affects would be captured from forest extent maps alone. But perhaps I misunderstand and roads/river allowing access to hunting are not important?

Yes you are correct, in response to this comment and comment No. 2 by Reviewer 2, we have re-run the analysis to include major roads and major and minor roads combined. However the accuracy of the road maps for the region is questionable, and the degree to which hunters and trappers have access to the forest via minor roads is impossible for us to ascertain (e.g., because these roads can degrade quickly). The new analysis in the supplementary information highlights this and we feel that our analysis represents the best approximation that we can currently make based on the data available. Discussion of this has been included in the revised manuscript.

Lines 225-244: While our method is a rapid and straightforward way for assessing population declines, there are several limitations that must be noted. In assessing the impact of exploitation, detailed information on the behaviour of wild bird trappers and species responses to exploitation is not available. Hence, we made simplified assumptions regarding hunting impact and accessibility, but in reality, species responses to hunting are more nuanced than the three categories we used (low, medium and high) and accessibility is a complex interaction of population, roads, topography and markets. We attempted to account for this uncertainty by combining maps of major roads and all available roads to calculate additional metrics of accessibility. Using road maps instead

of forest edges made a considerable difference to our results, with both road maps leading to considerably lower threat estimates.

Using only major roads only 16 species were above Red List thresholds (6 EN and 10VU) and with all roads 38 (3 CR, 13 EN, 22VU) (see supplementary figures 4 and 5). However, the maps we used, while the best available (open street map and WRI produced Indonesia map), are fundamentally inaccurate with many roads missing⁴³. Crucially, this inaccuracy is not uniform and changes the analysis in biased and unpredictable ways. For example, the maps contain no roads inside protected areas in Java, suggesting Javanese forests are much more isolated than they actually are, leading to large underestimations of threat. Secondly, deciding to what extent individual roads are accessible to hunters requires further assumptions. By using distance from forest edge we avoid using maps that are known to be inaccurate⁴³ and we can account for access from other means (such as rivers) by assuming deforestation follows these points of access.

(6) The 77 species identified as important for trade: were market prices available for all of them?

No, unfortunately not. Our categories are broadly based on observations from markets and the currently available literature. Market prices were not used.

(7) Can the authors provide more information on the parameter used in the PERT distribution modelling?

The PERT distribution was used to assess the uncertainty of our expert estimates in hunting accuracy. Our experts provided three estimates based on their own knowledge of the region; the maximum decline, minimum decline and likely decline. We then generated a PERT distribution using the conservative shape parameter of 4 (this dictates the slope of the lines which makes the distribution) and sampled the confidence intervals based on that.

Lines 417-423: F.E.R. and D.P.E. assigned an expected, maximum and minimum exploitation pressure (efficacy) at 5 km to these three categories to reflect their uncertainty in the categories given. We used these estimates to parameterize PERT (programme evaluation and review technique) distributions (shape = 4). PERT distributions assign low probabilities to extreme values and thus are useful for assessing the uncertainty in expert estimates. In our case, we assigned a minimum, expected, and maximum exploitation impact estimate and used the 2.5% and 97.5% percentiles as 95% uncertainty ranges.

(8) I am also not sure why the authors assessed trapping as an issue in Indonesia only?

We did this because for the majority of the species in our study, trapping is much more of a problem in Indonesia, as it is driven by the domestic trade in birds for the Javanese (largely) market. However, hunting and trapping does still occur in the other countries but at a much lower level. Assuming only Indonesia in our analysis is intended to make the analysis conservative, by reducing the risk of over estimation of trapping pressure in Malaysia, Singapore and Brunei. For some species which are hunted/trapped extensively outside of Indonesia (eg. Straw-headed bulbul) we have likely underestimated the impacts of exploitation, this has now been clarified in the supplementary tables. See also our more detailed response to Point 4 Reviewer 3 (see above).

(9) I can understand that the assumptions made (e.g. 5 km distance, market prices) might only be available for Indonesia (although this is not clear from the Methods. But the analyses are

implemented for other countries as well, in which hunting does occur. So by only using the method for one region, how does that allow you to compare across regions?

We find this comment confusing because we are not comparing across 'regions' at any point in the paper. Perhaps the reviewer means 'how does it allow us to compare across countries' instead. We actually do not compare the impacts of exploitation between countries, because there simply isn't any empirical data allowing us to do this. We know from studies of some species that exploitation is widespread throughout the region, and that for many species Indonesia is the primary issue.

If on the other hand, the reviewer is asking how generalisable our results are to other regions globally, the answer is somewhat. We believe our methods are potentially useful for estimating impacts on commercially valuable species in other regions, if they are calibrated to local contexts (eg hunter behaviour, habitat availability). We cannot say to what extent the combined impacts of habitat loss and wildlife trade are underestimated in other regions, but if our results are indicative of a global trend then the potential implications for global biodiversity are extremely negative.

(10) And if the authors deem that approach too risky for countries outside Indonesia, why not focus their study on Indonesia only?

We have answered this comment earlier in this review (see point 4, Reviewer 3)

(11, was 7 in the original review) The assumption of 1% population decline with 1% forest area decline across the range: is that a reasonable assumption given the many factors at play at this scale? Or perhaps the authors have evidence for that assumption?

The issue of linearity was also raised by review 2 (point 3), the answer is quoted below:

The Reviewer raises a good point and highlights a challenge for our analysis. The non-linearity and unpredictability of population responses to habitat loss pose a challenge for conservation policy makers and practitioners. While critical thresholds for habitat loss exist, they are very difficult and data intensive to calculate, and, for this reason, while the concept is ecologically interesting, it is of questionable utility to conservation decisions (as Swift et al 2009 attest). As to the question of time lags in population responses, these again are not especially useful to conservation practitioners, as from a management perspective (except in the case of a restoration project), a committed decline after a certain time lag can be considered the same as a decline that has already happened (REF?). For these reasons, we feel that the assumption of linearity in this case is justified as it is the translation of land cover change into declines that is most useful to conservation practitioners. Furthermore, the assumption of linearity is consistent with the application of the precautionary principle, a concept central to the Convention on Biological Diversity.

Secondly, we would like to thank the Reviewer for highlighting the issue of extinction debt. However we respectfully disagree with its implications for our results. Extinction debt refers to populations of species in remnant habitat patches that are already committed to extinction based on edge effects and inbreeding depression even though they remain in existence for a short additional time. By not considering this process, we have included many areas where populations are already committed to extinction, rendering our results more conservative.

On the other hand, we agree that the potential for differing species responses to habitat loss was not explained sufficiently in the manuscript and that our assumption of linearity needs to be justified. To address this issue, we have added a rigorous explanation (lines 272-287) to the Discussion. We are confident that these changes should satisfy the Reviewer's concerns.

Lines 272-287: We also assume a linear relationship between deforestation and population decline, which we believe is a conservative assumption given the negative impacts that edge, area and isolation effects have on species⁴⁷. There is considerable debate as to whether this relationship is linear, with many theoretical (and some field) studies suggesting that populations can remain reasonably stable until a certain threshold of habitat loss (for a comprehensive review see⁴⁸), leading to overestimates of population decline under linear assumptions. However, while habitat thresholds may exist for some species in our study, given our current knowledge, calculating these thresholds accurately is impossible and as such they cannot be incorporated and would be of questionable utility to conservation decisions⁴⁸. Instead, our assumption of linearity is in line with the precautionary principle (which is acknowledged in the Convention of Biological Diversity), since it ensures we are not underestimating declines by assuming a threshold that does not exist, and thus is more useful to conservation decisions. For simplicity, we also assumed that the effects of the loss of habitat from deforestation and exploitation were additive, which results in three species having maximum estimated declines above 100%. In reality, the proportion of the remaining area of a species subjected to exploitation will increase as the habitat is reduced and fragmented, meaning the impacts are likely synergistic⁴⁹, making our estimates for most species conservative.

Further Detailed Comments:

Line 29 Remote 'drives'

Changed

Line 47/48 I don't understand the logic behind the inclusion of this statement

We feel this statement identifies relevant similar literature and frames the gap in our knowledge that this study addresses.

Line 68: in the methods you imply that trapping as a factor for population declines was only estimated for Indonesia, but this is not obvious here. Furthermore, the statement might be confusing: so if trapping typically extends beyond 5 km, why only go up to 5 km?

This sentence is very briefly describing the nature of the results contained within the paragraph: we calculated the distance to edge across the whole range of the species in our study area, as explained in detail in the methods. The 5km issue is a reasonable assumption and has been covered in other responses (e.g., No 2 Reviewer 2).

Line 77: is that all 77 species?

Apologies, but we did not understand the question. Not all of the 77 regionally endemic species are endemic to Java, which is a small part of Sundaland, and we list all the species in question on the lines immediately following line 77.

Lines 82 ff: This is just wondering aloud: the species you classified as commercially valuable - were these all from Indonesia? And the areas in which you included trapping as a factor: was that only Indonesia? So if you find strong population declines, is this because your estimates are biased towards Indonesia?

There is only 1 commercially valuable species not found in Indonesia as pointed out earlier (e.g., No 3, reviewer 3). We calculated our trapping impacts by adjusting the proportion of the whole range of

the species that was 5km from the edge by the proportion in Indonesia. We did not apply trapping impacts in certain areas only. We do not understand what the Reviewer means by “biased towards Indonesia”. The majority of the land mass in Sundaland is in Indonesia, but we do not believe this creates a bias per se.

Line 360 Please explain that criterion

We agree more detail is required and have inserted a reference and the summary text of the criteria for further information. It now reads:

Line 441-447: We used Red List criterion A4 (an observed, estimated, inferred, projected or suspected population reduction where the time period must include both the past and the future (up to a max. of 100 years in future), and where the causes of reduction may not have ceased OR may not be understood OR may not be reversible), using the generation times provided by BirdLife International and the species-specific rate of habitat loss, to calculate the expected population decline over three generations or 10 years (whichever was longer).

Line 374 As someone not incredibly familiar with the intricacies of the Red List criteria and assessments, could the authors please provide more information on this section. Are the species reclassified from any of the classes that they were grouped into following the population decline assessment? How many species were reclassified? Which ones?

We can only recommend reclassification. Whether or not species are actually reclassified is up to Birdlife international / IUCN. However, we hope this study will generate significant discussion in Birdlife regarding reclassification. The number of species is in the text (please see Lines XXX) and the recommendations in full are contained in the supplementary tables X and Y.

Lines 385/386: did you focus on Indonesia only for this (similar to your approach above?) What data did you use to delineate protected areas.

This was the whole area not just restricted to Indonesia and we used the WDPa data. This clarification has been added to the text, which now reads (Lines 473-475): To determine the amount of legal protection species are presently afforded in Sundaland, we calculated the ESH of each species for 2015 that fell within a protected area (PA) of IUCN category I-V based on data from the world database of protected areas⁶⁸.

Please feel free to ask me for more details on some of my comments if they are unclear.

REVIEWERS' COMMENTS:

Reviewer #1 (Remarks to the Author):

The paper is much improved. I appreciate how attentive the authors were to my comments. I have no further suggestions.

Reviewer #2 (Remarks to the Author):

I am pleased to see that the revised version of manuscript by Symes and colleagues has improved substantially compared to the previous version. I think the authors have addressed most of my comments and the manuscript is now clearer and more substantiated. I especially appreciate the effort that the authors invested on defining different types of hunting-related threats for each species. Tables 1 and 2 in the Supplementary Material are extremely valuable resources that can be used by other authors in their research endeavours.

I still have several minor remarks:

Line 127-128: I would delete this sentence, it does not link up well with the rest of the paragraph. I mean, you mention extinction crisis is being more severe than anticipated, and then you present results for PA coverage of species ranges.

Line 138: 5.6% of what? Of the total study area? Of the combined species ranges?

Line 190: I don't think the authors can conclude that PAs play a critical role as reservoirs for the studied species because they have only looked at the amount of overlap between species ranges and PAs. However it would be interesting to look at whether habitat loss and (for some species) exploitation are larger or not inside and outside PAs for each species. In light of those results the authors could point out the importance of PAs for threatened species in the region.

Lines 190-200: I think the recent paper by Alice Hughes would contribute to the discussion here (Hughes, A. C. (2018). Have Indo-Malaysian forests reached the end of the road?. Biological Conservation, 223, 129-137.)

Lines 211-212: This has been addressed already in lines 198-200. I would avoid repetition and/or try to merge the information in this paragraph with the previous paragraph.

Lines 214- 218: Regarding the sentence: "these figures are exclusive of the additional extinction risks posed by hunting...", do you mean your estimate (16.9%) or the estimates by previous papers (24-42%)? Because I can imagine that the previous papers did not account for hunting and yet they report substantially higher estimates. I think this deserves further explanation.

Lines 231-244: The analysis using roads as accessibility points should be properly described in the methods section.

Line 238: Again, you could cite Hughes (2018) here. She mapped "unmapped" roads in the Indo-Malaysian region, and reported that, depending on the locality, between 23-100% of the roads are currently unmapped in the region.

Line 364: Here and in other instances, I would add the references in the text before the numbers so that the sentences are complete.

Lines 498-409: How did you take into account changes in elevation when calculating the distance to the closest forest edge?

Figure 3: In the legend, shouldn't it be between 0-5 km from the forest edge instead of 1-5?

Figure 4: How can error bars go over 100%? For example, for the wrinkled hornbill.

Figure 5: Please change categories to I-V to be consistent with lines 474-475.

Reviewer #3 (Remarks to the Author):

I am happy with the changes made and the response to my admittedly many comments.

Can you please check on Line 138 whether there are some words missing? Do you refer to 5.6 % of forest habitat if so, say so.

In brief: The authors made a great effort to respond to my concerns. The revised draft reads very well. The study makes an excellent point highlighting the need to account for both habitat loss and hunting/extraction pressure when assessing species responses to land use changes. There are limitations due to the lack of data in tropical regions and the authors are very clear on these limitations and their implications for the findings in this study. I look forward to seeing this paper published.

Marion Pfeifer

REVIEWERS' COMMENTS:

Reviewer #1 (Remarks to the Author):

Comment

The paper is much improved. I appreciate how attentive the authors were to my comments. I have no further suggestions.

Response

We are thankful to the reviewer for the helpful comments.

Reviewer #2 (Remarks to the Author):

Comment

I am pleased to see that the revised version of manuscript by Symes and colleagues has improved substantially compared to the previous version. I think the authors have addressed most of my comments and the manuscript is now clearer and more substantiated. I especially appreciate the effort that the authors invested on defining different types of hunting-related threats for each species. Tables 1 and 2 in the Supplementary Material are extremely valuable resources that can be used by other authors in their research endeavours.

Response

Thanks for the positive comments. We are thankful to the reviewer for helping us improve the manuscript.

Comment

I still have several minor remarks:

Line 127-128: I would delete this sentence, it does not link up well with the rest of the paragraph. I mean, you mention extinction crisis is being more severe than anticipated, and then you present results for PA coverage of species ranges.

Response

We deleted the sentence.

Comment

Line 138: 5.6% of what? Of the total study area? Of the combined species ranges?

Response

It refers to the species' extent of suitable habitat. We have clarified this now.

Comment

Line 190: I don't think the authors can conclude that PAs play a critical role as reservoirs for the studied species because they have only looked at the amount of overlap between species ranges and PAs. However it would be interesting to look at whether habitat loss and (for some species) exploitation are larger or not inside and outside PAs for each species. In light of those results the authors could point out the importance of PAs for threatened species in the region.

Response

It is true that we do not know whether PAs reduce bird exploitation. We added a line to the discussion to note that this would be an important area of future research.

Action

The text was modified as follows:

“Our analysis suggests that exploitation for wildlife trade has caused dramatic declines in many species within the region, and underscores the critical role that effectively guarded PAs ~~can~~ could play as reservoirs of these species. It remains poorly unknown, however, whether PAs are effective at reducing bird exploitation on the ground and future research should point in this direction.”

Comment

Lines 190-200: I think the recent paper by Alice Hughes would contribute to the discussion here (Hughes, A. C. (2018). Have Indo-Malaysian forests reached the end of the road?. Biological Conservation, 223, 129-137.)

Response

Thanks for pointing this out. We have added a citation for Hughes (2018) and commented on her findings in the discussion.

Comment

Lines 211-212: This has been addressed already in lines 198-200. I would avoid repetition and/or try to merge the information in this paragraph with the previous paragraph.

Response

We agree and have removed this sentence to avoid repetition.

Comment

Lines 214- 218: Regarding the sentence: “these figures are exclusive of the additional extinction risks posed by hunting...”, do you mean your estimate (16.9%) or the estimates by previous papers (24-42%)? Because I can imagine that the previous papers did not account for hunting and yet they report substantially higher estimates. I think this deserves further explanation.

Response

These are our estimates. We have clarified this in the text. Our estimates for the SAR are based on only habitat loss. We have replaced the word “exclusive” with “do not include” to avoid confusion.

Comment

Lines 231-244: The analysis using roads as accessibility points should be properly described in the methods section.

Response

We thank the reviewer for pointing this out, the methodology used was exactly the same we just recalculated the path distance rasters to measure the distance from the nearest road rather than forest edge. We have added the following to the methods (lines 390-395):

To test the sensitivity of our result to different access points we also created path distance rasters based on the road network in the region obtained from OpenStreetMap (for Malaysia, Singapore and Brunei)⁶⁴ and the Peta Dasar (for Indonesia)⁶⁵. We then calculated the proportion of the species ranges that was both within 5km of a road and inside forest from the path distance rasters. We repeated this analysis twice, first for only major roads and again including all roads in the maps.

Comment

Line 238: Again, you could cite Hughes (2018) here. She mapped “unmapped” roads in the Indo-Malaysian region, and reported that, depending on the locality, between 23-100% of the roads are currently unmapped in the region.

Response

We agree and have added the citation.

Comment

Line 364: Here and in other instances, I would add the references in the text before the numbers so that the sentences are complete.

Response

We added the names of the authors as suggested.

Comment

Lines 498-409: How did you take into account changes in elevation when calculating the distance to the closest forest edge?

Response

This was done using the Path Distance tool in ArcGIS as noted in the methods.

Comment

Figure 3: In the legend, shouldn't it be between 0-5 km from the forest edge instead of 1-5?

Response

Thanks for noting this. We corrected this error.

Comment

Figure 4: How can error bars go over 100%? For example, for the wrinkled hornbill.

Response

This results from adding the effects of habitat loss and exploitation separately. We prefer to leave this numbers above 100% to denote a higher certainty of high pressure. We have added now a note to the figure legend to clarify.

Action

The following text was added to the legend:

“Values above 100% result from adding the effects of habitat loss and exploitation and are interpreted as population declines of 100%.”

Comment

Figure 5: Please change categories to I-V to be consistent with lines 474-475.

Response

Thanks for noting this, we made this change.

Reviewer #3 (Remarks to the Author):

Comment

I am happy with the changes made and the response to my admittedly many comments.

Response

We thank the reviewer for helping us improve the manuscript.

Comment

Can you please check on Line 138 whether there are some words missing? Do you refer to 5.6% of forest habitat if so, say so.

Response

It refers to the species' extent of suitable habitat. We have clarified this now in the text.

Comment

In brief: The authors made a great effort to respond to my concerns. The revised draft reads very well. The study makes an excellent point highlighting the need to account for both habitat loss and hunting/extraction pressure when assessing species responses to land use changes. There are limitations due to the lack of data in tropical regions and the authors are very clear on these limitations and their implications for the findings in this study. I look forward to seeing this paper published.

Marion Pfeifer

Response

Again, many thanks for your help and the encouraging comments.